# GenCtrl – A Formal Controllability Toolkit for Generative Models

**Emily Cheng**[*,1,2]**, Carmen Amo Alonso**[3]**, Federico Danieli**[1]**, Arno Blaas**[1]**,**
**Luca Zappella**[1]**, Pau Rodríguez**[1]**, Xavier Suau**[1]
[1]Apple, [2]Universitat Pompeu Fabra, [3]Stanford

## Abstract

As generative models become ubiquitous, there is a critical need for fine-grained control over the generation process. Yet, while controlled generation methods from prompting to fine-tuning proliferate, a fundamental question remains unanswered: are these models truly controllable in the first place? In this work, we provide a theoretical framework to formally answer this question. Framing human-model interaction as a control process, we propose a novel algorithm to estimate the controllable sets of models in a dialogue setting. Notably, we provide formal guarantees on the estimation error as a function of sample complexity: we derive probably-approximately correct bounds for controllable set estimates that are distribution-free, employ no assumptions except for output boundedness, and work for any black-box nonlinear control system (*i.e.,* any generative model). We empirically demonstrate the theoretical framework on different tasks in controlling dialogue processes, for both language models and text-to-image generation. Our results show that model controllability is surprisingly fragile and highly dependent on the experimental setting. This highlights the need for rigorous controllability analysis, shifting the focus from simply attempting control to first understanding its fundamental limits.

## 1 Introduction

The widespread deployment of generative models has driven significant research effort into controlling their outputs (Zhang et al., 2023). A diverse array of controlled generation methods has been developed, from prompting (Marvin et al., 2023) and finetuning (Wei et al., 2022a; Ouyang et al., 2022b) to steering towards specific styles or concepts (Li et al., 2024; Rodriguez et al., 2025b; Rimsky et al., 2024; Wu et al., 2024). However, beneath the surface of this empirical progress lies a set of foundational, yet largely unexamined, assumptions. Specifically, current methods implicitly assume the model is fundamentally controllable. This rests on three key premises. First, they assume **reachability**: that a desired set of outputs is achievable using a given control mechanism and initial prompt. Second, they assume universal **controllability**, meaning the desired output is reachable from any initial state. Finally, they assume **calibration**, that is, the output is a direct function of the control variable. The absence of tools to formally verify these assumptions casts uncertainty over the reliability and fundamental limits of current controlled generation techniques.

In this work, we bring these assumptions to the forefront. We provide tools to verify them by turning to control theory (Sontag, 1998), adapting its methods to formalize the problem for modern AI systems. Casting controlled generation as a *control process*, we propose a framework to quantify its reachability and controllability with probabilistic guarantees. Our framework treats a generative model as a black-box nonlinear control system; it is not only agnostic to model architecture but is also designed to handle discrete or continuous-valued input and output spaces (*e.g.,* textual prompts and generations). Finally, we demonstrate our framework on a user-model dialogue setting, a common use-case of current generative models (Google, 2024; Yang et al., 2025; OpenAI, 2023).

Our primary contributions are as follows:

---

[*]Correspondence to: `emilyshana.cheng@upf.edu`

- **A formal framework for controllability.** We introduce a control-theoretic framework to rigorously define and quantify the reachable and controllable sets for any black-box system, including generative models. To the best of our knowledge, this provides the first formal language to characterize the operational boundaries of generative model control.
- **PAC algorithms for controllable set estimation.** We present Monte Carlo algorithms for reachable and controllable set estimation from black-box system interactions, along with probably-approximately correct (PAC) confidence bounds (Thm. 1 and 2). The algorithms are particularly well-suited for generative models as they are distribution-free, only assume boundedness of the target attribute, and explicitly account for the discrete bottlenecks inherent in many generative systems.
- **Empirical findings.** We conduct a broad empirical analysis of controllability across models and tasks for text generation with LLMs, and image generation with text-to-image models (T2IMs). Leveraging our theoretical results (Thm. 1 and 2), our experiments reveal that model controllability is not guaranteed. The significant heterogeneity of results across modalities, models, and tasks signals the need for case-specific controllability analyses.
- **An open-source toolkit.** We release our framework and algorithms in a PyTorch library at `https://github.com/apple/ml-genctrl` to facilitate analysis of controllability by the broader research community.

This work argues for a paradigm shift where the controllability of generative models is not an implicit assumption but an explicit subject of analysis. By providing formal tools to do so, we lay a more principled foundation for future work in controllable AI.

## 2 RELATED WORK

**Controlling generative models**  Research on controlling generative model outputs follows three main paradigms: **(i)** Prompt engineering, *e.g.,* in-context learning or chain-of-thought prompting (Brown et al., 2020; Wei et al., 2022b); **(ii)** Finetuning, *e.g.,* via RLHF or DPO (Ouyang et al., 2022a; Rafailov et al., 2023), to align the model behavior; **(iii)** Representation engineering, which directly manipulates model activations (Turner et al., 2023; Li et al., 2024; Suau et al., 2024; Rodriguez et al., 2025a; Wu et al., 2024; Cheng et al., 2024). Our framework can assess any control mechanism including all examples above. In this work we mainly focus on prompting, given its widespread use.

**Control theory and reachability analysis for generative models**  Controllability and reachability are core concepts in control and dynamical systems theory (Sontag, 1998; Borrelli et al., 2017). Despite their widespread use in other areas of engineering, such as robotics (Nakamura et al., 2025), the application of control theoretic concepts to machine learning has mainly been limited to reinforcement learning (Recht, 2019), or to analyze training dynamics (Han et al., 2019). Despite recent efforts to develop theory for generative models (Soatto et al., 2023; Marchi et al., 2024), these often rely on impractical assumptions. Extending control-theoretic concepts to large generative models poses several challenges, mostly due to their nonlinear and high-dimensional nature (Bansal et al., 2017). In the control literature, past works have addressed model-free reachable set estimation by proposing data-driven methods. However, existing methods do not apply to generative model outputs for three reasons: either they **(i)** apply to *state* and not output reachability (Devonport et al., 2021; Devonport and Arcak, 2020; Dietrich et al., 2024); **(ii)** impose assumptions, *e.g.,* Lipschitzness, on system dynamics that are not verifiable for black-box LLMs and T2IMs (Choi et al., 2025; Park et al., 2024; Xue et al., 2020); **(iii)** return continuous reachable set estimates while LLMs and T2IMs' reachable sets are countable, making the estimates vacuously large (Devonport and Arcak, 2019; Sivaramakrishnan et al., 2025; Devonport and Arcak, 2020; Dietrich et al., 2024). We bridge these gaps by developing a black-box sampling algorithm that estimates non-vacuous reachable and controllable sets for generative models, with probabilistic guarantees. This paves the way for rigorous control-theoretic analysis viable even for modern, opaque large-scale models.

## 3 PROBLEM FORMULATION

We study control of LLMs or T2IMs in a dialogue setting. In Sec. 3.1 we formalize the dialogue process through a control-theoretic lens (Sontag, 1998) and in Sec. 3.2 we define reachability and controllability under this dialogue formalization.

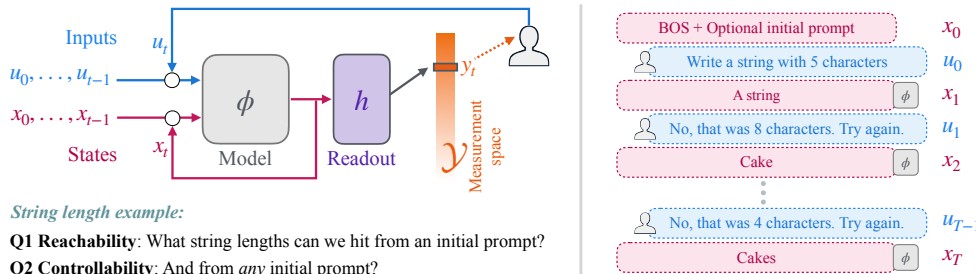

Figure 1: **Dialogue Process.** (Left) Schema of a dialogue process as a control process, showing the roles of each of the concepts introduced in Sec. 3.1. Theorems 1 and 2 tell how many inputs and initial states (respectively) should be sampled to answer **Q1** and **Q2** with confidence $\delta$. (Right) Example of dialogue process.

## 3.1 DIALOGUE WITH A GENERATIVE MODEL AS A CONTROL PROCESS

We study a *dialogue process* (DP) with generative models, *i.e.,* LLMs and T2IMs (Fig. 1). Next, we describe the problem setup, which we formalize in the language of dynamical systems theory.

**Time Domain** The time domain $\mathcal{T}$ is the set that indexes the system's temporal evolution. For a DP, $\mathcal{T} = \mathbb{N}$ is discrete, where each $t \in \mathbb{N}$ corresponds to the $t^{th}$ turn in the dialogue process.

**State Space** The state space $\mathcal{X}$ is the set of possible states the system can occupy. Each $x \in \mathcal{X}$ describes the system at a given time; for $t = 0$, $\mathcal{X}$ is the space of initial string contexts, and for $t \geq 1$, $\mathcal{X}$ is the space of possible generations by the model. In a DP with an LLM or T2IM, $\mathcal{X}$ is the set of strings and of strings and images, respectively.

**Control Input Space** The control input space $\mathcal{U}$ is the set of all possible inputs that can intervene the system. Each control input $u \in \mathcal{U}$ denotes an intervention to the system at a given time. In practice, $u$ is any degree of freedom tuned by the user or system designer to affect generation, *e.g.,* a prompt, or activation steering (Rodriguez et al., 2025b; Li et al., 2024). While $\mathcal{U}$ can be any type of intervention, we *focus on prompting as the primary way users interact with generative models*.

**Dynamics Map** We frame dialogue as a control process where user inputs provide feedback to guide the model's behavior. In particular, we treat these inputs as interventions to the generative model. An *intervened model* $\phi$ maps prompts and any other control variables to the next model generation, defining dynamics $\phi : \mathcal{X} \times \mathcal{U} \to \mathcal{X}$, where $x_{t+1} = \phi(x_t, \ldots, x_0; u_t, \ldots, u_0)$. [1]

Controlled generation aims to achieve a desired behavior as quantified by a metric (*e.g.,* a sentiment score). This is formalized via a measurement value and readout map, defined below.

**Measurement Value** The measurement-value space $\mathcal{Y}$ is the set of possible measurable outputs of the system. In particular, the measurement value is how we evaluate the system's behavior at a given time. This may correspond to external assessments of the model's generation, *e.g.,* sentence length or number of cars in an image. The choice of $\mathcal{Y}$ is task-dependent. Controlled generation, through input interventions, aims to restrict the system $\phi$ measurements to a desired subset $\mathcal{Y}' \subset \mathcal{Y}$.

**Readout Map** The readout map $h : \mathcal{T} \times \mathcal{X} \to \mathcal{Y}$ maps the state at time $t$ to a measurement value. For DPs, $h(t, x_t) =: h(x_t)$ is given by a deterministic function that maps model generations $x_t$ to measurement values in $\mathcal{Y}$. For example, if the attribute being controlled is the LLM's output string length, the readout $h$ may be the Python `len()` function.

The above concepts permit a rigorous definition of a dialogue process between a user and intervened model $\phi$ as a discrete-time control system (Def. 1). As shown in Fig. 1, the user and model take turns in a conversation starting from the user's initial prompt. Subsequent user prompts are interventions that trigger another dialogue turn after the model's response.

---

[1]In practice, LLMs come with a *sampling strategy* for extracting $g$ from $\phi$, *e.g.,* greedy decoding or top-$p$ sampling. Our definition of $\phi$ abstracts away the sampling strategy, which is already reflected in $x_{t+1}$.

**Definition 1.** A *dialogue process* is a stochastic discrete-time, nonlinear control system parametrized by the tuple $(\phi, \mathcal{T}, \mathcal{X}, \mathcal{U}, h, \mathcal{Y})$, such that:

$$x_0 = \text{initial prompt} \in \mathcal{X}, \tag{1a}$$

$$x_t = \phi(x_{t-1}, \ldots, x_0; u_{t-1}, \ldots, u_0) \in \mathcal{X}, \tag{1b}$$

$$y_t = h(x_t) \in \mathcal{Y}, \tag{1c}$$

for $t \in \mathbb{N}$, where $u_{0\ldots t-1}$ are a collection of inputs from $\mathcal{U}$.

## 3.2 Reachability & Controllability in Dialogue Processes

Definition 1 establishes a dialogue process as a control process. In this work, we do not study the question of how to choose an input from $\mathcal{U}$ to elicit some result (controller design). Instead, we tackle the more fundamental question of *what is possible* to achieve over a time horizon $T$ **(i)** given a specific initial state, *i.e.,* a prompt $x_0$; and **(ii)** for any or all initial states. These two are well-studied in control theory: **(i)** pertains to the question of *reachability*, while **(ii)** addresses *controllability*. In what follows, we formalize these definitions in the context of generative models.

**Definition 2.** Given a dialogue process (Def. 1) with an initial state $x_0 \in \mathcal{X}$, then the *reachable set* of outputs for the model with dynamics $\phi$ is given by

$$\mathcal{R}(x_0, \mathcal{U}, t) = \{\tilde{y} \in \mathcal{Y} \mid \exists\, u_0, \ldots, u_{t-1} \in \mathcal{U} \text{ and } y_t = \tilde{y}\}. \tag{2}$$

That is, given an initial state $x_0 \in \mathcal{X}$, the reachable output set at time $t$ is the set of all $y \in \mathcal{Y}$ reached when integrating the dynamics $t$ steps, with some control sequence $u_0, \ldots, u_{t-1} \in \mathcal{U}$. Def. 2 relies on a fixed initial state $x_0$. It is then natural to ask what is reachable from all initial states. Namely, if all $x_0 \in \mathcal{X}$ can reach all values in the set $\mathcal{Y}' \subseteq \mathcal{Y}$ using some input sequence in $\mathcal{U}$, then the system is *controllable* on $\mathcal{Y}'$ (Sontag, 1998). Formally,

**Definition 3.** A system is *controllable* on $\mathcal{Y}' \subseteq \mathcal{Y}$ if $\exists t \geq 0$ such that $\mathcal{R}(x_0, \mathcal{U}, t) = \mathcal{Y}' \,\forall\, x_0 \in \mathcal{X}$.

A differentiating feature of LLMs and T2IMs is the discrete nature of string prompts, which makes the reachable set countable (see App. E for discussion and formal proofs). We refer to this as the **discrete bottleneck**. This property affects both reachability (Def. 2) and controllability (Def. 3), which are preconditions for system *control*.

In Sec. 4 we propose *coarse-grained* reachability as a relaxation of the problem for DPs with a discrete bottleneck. In Sec. 4.1 we derive a PAC bound to approximate the reachable set of a given initial state, then in Sec. 5 present a PAC bound to estimate the controllable set of the system. Finally, in Sec. 6 we use our theory to estimate reachable and controllable sets of current LLMs and T2IMs, finding they are seldom controllable on simple tasks.

## 4 Monte Carlo Reachability for Dialogue Processes

Generative models are nonlinear, high-dimensional, and have unknown dynamics. Therefore, it is nontrivial to analytically derive the reachable set. Hence, we rely on *statistical guarantees* that only depend on the system's empirical trajectories. In Sec. 4.1 we tailor the definition of reachability to the discrete bottleneck of LLMs and T2IMs. Then, we propose a sample complexity bound for the reachable set in Sec. 4.2, which we use in Sec. 4.3 in an algorithm that computes an error-bounded reachable set approximation. Since reachable set estimates will rely on sampling, we first set up a *probabilistic notion* of the control process, as in prior work (Devonport and Arcak, 2019):

- $X_0 \sim p_0$, where $p_0 : \mathcal{X}_0 \to [0, 1]$ is a probability density over the set of initial states $\mathcal{X}_0 \subset \mathcal{X}$.
- $U_t \sim p_{u,t}$, where $p_{u,t} : \mathcal{U} \to [0, 1]$ is a probability density over the input space $\mathcal{U}$ at time $t \geq 0$.
- $X_t = \phi(t; X_0, U_{0\ldots t-1}) \sim p_t$, where $p_t : \mathcal{X}_t \to [0, 1]$ is the pushforward of $p_0$ and $p_u$ under system dynamics, for $t \geq 0$.
- $Y_t = h(X_t, U_t) \sim p_{y,t}$, where $p_{y,t} : \mathcal{Y} \to [0, 1]$ is the probability density over measurements at time $t \geq 0$.

### 4.1 COARSE-GRAINED REACHABILITY OVERCOMES THE DISCRETE BOTTLENECK

The discrete bottleneck in DPs impacts the reachable set: even if the measurement value is *continuous-valued* (*e.g.,* continuous text formality score), the true reachable set can be countable. This means we cannot directly apply existing reachable set estimators that return *continuous sets* (Devonport and Arcak, 2019; Alanwar et al., 2021), as the returned sets would be vacuously large.

A workaround is coarse-grained reachability, a simplified but faithful discrete abstraction of the original measurements (Ren and Dimarogonas, 2020). We consider two cases: **(i)** categorical measurements, *e.g.,* $\{\text{toxic}, \text{non-toxic}\}$, which require no abstraction (they are already discrete); **(ii)** continuous-valued measurements, *e.g.,* a formality score in $[0, 1]$. For the latter, we relax the reachability problem to a $\gamma$-quantized one: instead of exactly reaching values in $\mathcal{Y}$, we allow an error margin $\gamma$ (Def. 4). For instance, rather than requiring $0.\overline{3}$ exactly, we aim for $0.\overline{3} \pm 0.05$.

**Definition 4** ($\gamma$-quantized Reachable Set). Let $(\phi, \mathcal{T}, \mathcal{X}, \mathcal{U}, h, \mathcal{Y})$ be a dialogue process per Def. 1. The $\gamma$-*quantized reachable output set* at time $t$ from $x_0$ is given by

$$\mathcal{R}^\gamma(x_0, \mathcal{U}, t) = \{\tilde{y} \in \mathcal{Y} \mid \exists\, u_0, \dots, u_{t-1} \in \mathcal{U} \text{ and } \|y_t - \tilde{y}\|_\infty \leq \gamma\}, \tag{3}$$

where $\| \cdot \|_\infty$ refers to the $\infty$-norm and $\gamma \in \mathbb{R}_+$.[2]

For continuous-valued measurements, our goal is now to estimate $\mathcal{R}^\gamma$. Recovering $\mathcal{R}^\gamma$ *guarantees* each point in $\mathcal{R}^\gamma$ is at most $\gamma$ away from some point in the true reachable set. For brevity, we refer to reachable set estimation over both categorical or quantized $\mathcal{Y}$ as coarse-grained reachability.

### 4.2 SAMPLE COMPLEXITY BOUND FOR COARSE-GRAINED REACHABLE SET ESTIMATION

Our goal is a PAC bound that estimates $\mathcal{R}^\gamma$. To build up to this bound, we first find a suitable quantization $\mathcal{Y}_q$ of the measurement-value space $\mathcal{Y}$, requiring that $\mathcal{Y}_q$ has finite cardinality, *i.e.,* that $\mathcal{Y}$ is bounded. If $\mathcal{Y}$ is already categorical, then $\mathcal{Y}_q = \mathcal{Y}$. If $\mathcal{Y}$ is continuous-valued, $\mathcal{Y}_q$ is a minimal cover of $\mathcal{Y}$ using $\infty$-balls of radius $\gamma/2$, where $\gamma$ is the user-defined error in Def. 4. Let $N = |\mathcal{Y}_q|$ be the number of bins in the discretization: $|\mathcal{Y}|$ if categorical and the covering number of $\mathcal{Y}$ otherwise. Since our algorithm will be based on sampling, we need a *probabilistic* notion of error, which we define using $\mathcal{Y}_q$. In particular, the user sets a small threshold $p \in (0, 1)$ that tunes the precision of the estimate. Then, to construct the *p-approximation* of $\mathcal{R}(x_0, \mathcal{U}, t)$ (hereon $\mathcal{R}_t$), we keep all points lying in the bins $y_{\text{bin}} \in \mathcal{Y}_q$ with probability mass $p_{y,t}(y_{\text{bin}}) \geq p$; any bin with density $< p$ is considered negligible and discarded. We formally define $p$-approximation below:

**Definition 5** (*p*-Approximate Measurement Value). Alternative and equivalent definitions are given for the categorical and $\gamma$-quantized cases.

- **Categorical:** The $(p)$-approximation of $\mathcal{R}_t$ is $\mathcal{R}_{t,p} := \{y \mid y \in \mathcal{Y},\ p_{y,t}(\{y\}) \geq p\}$.
- **$\gamma$-Quantized:** The $(p, \gamma)$-approximation of $\mathcal{R}_t$ is $\mathcal{R}_{t,p}^\gamma := \{y \mid \exists y_{\text{bin}} \in \mathcal{Y}_q : y \in y_{\text{bin}} \wedge p_{y,t}(y_{\text{bin}}) \geq p\}$, where $y_{\text{bin}} \in \mathcal{Y}_q$ is an $\infty$-ball of radius $\gamma/2$ in $\mathcal{Y}$'s cover.

Now, using $\mathcal{R}_{t,p}^{(\gamma)}$, we state the sample complexity bound. For brevity, we combine categorical and quantized versions of the result into the below Theorem (see App. F.1 for separate statements):

---

**Theorem 1: Sample complexity bound, coarse-grained reachability**

Let $\mathcal{Y}_q$ have finite cardinality $N$. Let $m$ be the number of i.i.d. samples drawn from $Y_t$, *i.e.,* $\{y_i\}_{i=1}^m$ with each $y_i \in \mathcal{Y}$ $\forall i$. Fix $\delta \in (0, 1)$. If

$$m \geq \max\left(N, \frac{\log(\delta/N)}{\log(1 - p)}\right), \tag{4}$$

then $\mathbb{P}(\mathcal{R}_{t,p}^{(\gamma)} \subset \hat{\mathcal{R}}_t) \geq 1 - \delta$, where $\hat{\mathcal{R}}_t = \{y_i\}_{i=1}^m$ if categorical, and $\hat{\mathcal{R}}_t = \cup_{i=1}^m \mathcal{B}_\infty(y_i, \gamma)$ if quantized ($\mathcal{B}_\infty(y_i, \gamma)$ is the $\infty$-ball centered at $y_i$ with radius $\gamma$). Proof in App. F.1.

---

[2]In practice, $\mathcal{Y} \in \mathbb{R}^n$ can consist of several unrelated attributes $1 \dots n$ like *formality* and *sentence length*. The $\infty$-norm makes the theory already hold for multiple orthogonal readout dimensions.

Theorem 1 shows that, after $m$ samples, one is confident that all reachable bins in $\mathcal{Y}_q$ have been covered; the confidence is given by $1 - \delta$, and the precision of the estimate by $p$. Then, if a target set $\mathcal{Y}^*$ is not included in $\hat{\mathcal{R}}_t$, that $\mathcal{Y}^*$ is unreachable with probability $\geq 1 - \delta$. By construction, $\hat{\mathcal{R}}_t$ is a *tight estimate* of the true $\mathcal{R}_t^{(\gamma)}$: all points in $\hat{\mathcal{R}}_t$ are guaranteed to lie within $\gamma$ of a reached point.

## 4.3 MONTE CARLO SAMPLING ALGORITHM FOR REACHABILITY IN DIALOGUE PROCESSES

Theorem 1 provides the foundation for a Monte Carlo algorithm to estimate a DP's reachable sets. The procedure to build the reachable set estimate $\hat{\mathcal{R}}_t$ is stated in Alg. 1a. It shares the same structure for both discrete and continuous cases with case-specific post-processing of the samples. Under sample size requirements from Thm. 1, Alg. 1a guarantees $\mathbb{P}(\mathcal{R}_{t,p}^\gamma \subset \hat{\mathcal{R}}_t^{(m)}) \geq 1 - \delta$. Crucially, this guarantee uses no information about the underlying model: the number of trajectories $m$ does not rely on whether the model is stochastic[3] nor the timestep. In App. F.4, we empirically validate the bound in Thm. 1. In App. F.3, we analyze $m$'s dependence on parameters, finding $m \sim O(N \log N)$.

## 5 MONTE CARLO CONTROLLABILITY FOR DIALOGUE PROCESSES

Having established reachable sets for individual initial states (Alg. 1a), we now quantify controllability of the system. Since controllability entails reaching a subset $\mathcal{Y}' \subset \mathcal{Y}$ from *all* initial states, one can naturally estimate controllable sets by intersecting the reachable sets of $k$ initial states. To measure how controllable the system is, we first define *partial controllability* (Sontag, 1998), then derive a PAC bound (Thm. 2) on the DP's partially controllable set. This yields Alg. 1b, an algorithm for controllable set estimation. As Alg. 1b will rely on reachable set estimates that are $p$-approximate (Def. 5), we port over the notion of $p$-approximate for controllability. Since we focus on the discrete bottleneck, we directly consider the discretization $\mathcal{Y}_q$ of the measurement space $\mathcal{Y}$.

To formalize *partial controllability*, we introduce a measure $\mu : \mathcal{Y}_q \to [0,1]$, where $\mu(y_{\text{bin}})$ quantifies the *true proportion* of initial states that $p$-approximately reach a $y_{\text{bin}}$ in $\mathcal{Y}_q$. Namely, $\mu : y_{\text{bin}} \mapsto \mathbb{P}_{x_0 \sim p_0}[y_{\text{bin}} \in \mathcal{R}_{t,p}^\gamma(x_0)]$. We use $\mu$ to define the *$\alpha$-controllable set* $\mathcal{C}_t^\alpha$: the set of bins in $\mathcal{Y}_q$ that are *$p$-approximately reached* by a large proportion ($\geq 1 - \alpha$) of initial states $x_0$. Formally,

**Definition 6** ($\alpha$-controllable set). Given a control system $(\phi, \mathcal{T}, \mathcal{X}, \mathcal{U}, h, \mathcal{Y})$, the $\alpha$-controllable set $\mathcal{C}_t^\alpha \subseteq \mathcal{Y}$ at time $t$ is given by $\mathcal{C}_t^\alpha = \{y_{\text{bin}} \mid \mu(y_{\text{bin}}) \geq 1 - \alpha\}$.

---

**Example 5.1: $\alpha$-controllable set**

Consider an initial state space $\{\texttt{BOS}, \text{Hi}, \text{Hello}, \text{Hey}\}$ with possible inputs "Generate a sentence of length $\{1 \cdots 10\}$." Say each outcome in $\texttt{length}=\{2, 3, 5, 8\}$ is $p$-approximately reached by exactly 3 states (75% of states, they don't have to be the same ones each time). Then, for $\alpha = 0.25$ (*i.e.,* 25%), the $\alpha$-controllable set $\mathcal{C}_t^\alpha$ (Def. 6) is $\texttt{length}=\{2, 3, 5, 8\}$.

---

We aim to estimate the true $\alpha$-controllable set $\mathcal{C}_t^\alpha$ by sampling $k$ initial states, sampling their reachable sets, and taking their intersection: $\hat{\mathcal{C}}_t = \cap_{i=1}^k \hat{\mathcal{R}}_t(x_0^{(i)})$. The key question is how many initial state samples $k$ are needed to approximate $\mathcal{C}_t^\alpha$ with high confidence. To answer the question, we need a notion of *approximation error* between the estimated $\hat{\mathcal{C}}_t$ and the target $\mathcal{C}_t^\alpha$. A natural choice is the measure under $\mu$ of *false positives* in $\hat{\mathcal{C}}_t$ with respect to $\mathcal{C}_t^\alpha$, formally, $\mu(\hat{\mathcal{C}}_t \backslash \mathcal{C}_t^\alpha)$.[4] With enough samples, we can bound $\mu(\hat{\mathcal{C}}_t \backslash \mathcal{C}_t^\alpha)$ below a desired $\epsilon \in (0,1)$ with high probability:

---

[3] For deterministic $\phi$, $\hat{\mathcal{R}}_t^{(m)}$ reflects *exact (or within-$\gamma$) controllability*, and $p$-approximation only captures sampling uncertainty. For stochastic $\phi$, $\hat{\mathcal{R}}_t^{(m)}$ reflects controllability *in-distribution*: $p$ reflects stochasticity of both sampling and $\phi$. See App. C for discussion.

[4] There are no false negatives — $\hat{\mathcal{C}}_t$ is a strict overapproximation of $\mathcal{C}_t^\alpha$, as $\hat{\mathcal{C}}_t \supset \mathcal{C}_t^\alpha$ by construction — taking successive intersections shrinks the running controllable set until convergence.

---

**Algorithm 1a:** Monte Carlo Reachability Estimation for Dialogue Processes

---

**Input:** Initial state $x_0$, confidence level, probability threshold $(\delta, p) \in (0,1)^2$
**Input:** If $\gamma$-quantized: quantization parameter $\gamma > 0$

1  Compute sample size $m$;      // Thm. 1
   // Sample generation
2  **for** $i = 1$ **to** $m$ **do**
3       Integrate DP to time $t$: $x_t = \phi_t(x_0, u)$ sampling input sequence $u \sim p_U$;
4       Evaluate output: $y_t = h(x_t, u_t)$;
5  **return** $\hat{\mathcal{R}}_t^{(m)} = \bigcup_{i=1}^m \mathcal{B}_\infty(y_t^{(i)}, \gamma)$ **if** $\gamma$-quantized reachability **else** $\{y_t^{(i)}\}_{i=1}^m$;

---

**Algorithm 1b:** Monte Carlo Controllability Estimation for Dialogue Process

---

**Input:** Confidence level, partial controllability, error, probability threshold $(\delta, \alpha, \epsilon, p) \in (0,1)^4$
**Input:** If $\gamma$-quantized: quantization parameter $\gamma > 0$

1  Compute initial state $k$, reachable set sample sizes $m$;      // Thm. 2, see App. H
   // Sample generation
2  **for** $i = 1$ **to** $k$ **do**
3       Sample initial state: $x_0^{(i)} \sim p_0$;
4       Sample reachable set: $\hat{\mathcal{R}}_t^{(m)}(x_0^{(i)})$; // Alg. 1a
5  **return** *Empirical controllable set:* $\hat{\mathcal{C}}_t = \bigcap_{i=1}^k \hat{\mathcal{R}}_t(x_0^{(i)})$;

---

**Algorithm 1**: Pseudocode for Monte Carlo reachability (left) and controllability (right) estimation.

---

**Theorem 2: Sample complexity bound, $\alpha$ controllable set**

Fix $\epsilon, \delta_C, p, \alpha \in (0,1)$ (and $\gamma \in \mathbb{R}_+$). Let $\delta_R$ be the confidence from Thm. 1. Given $k$ initial states $\{x_0^{(i)}\}_{i=1}^k \overset{\text{i.i.d.}}{\sim} p_0$ and their reachable set estimates $\{\hat{\mathcal{R}}_t(x_0^{(i)})\}_{i=1}^k$, if

$$k \geq \frac{\log \epsilon \delta_C}{\log(1-\alpha)}, \tag{5}$$

then $\mathbb{P}(\mu(\hat{\mathcal{C}}_t \setminus \mathcal{C}_t^\alpha) < \epsilon) \geq (1-\delta_C)(1-\delta_R)^k$, where $\hat{\mathcal{C}}_t = \bigcap_{i=1}^k \hat{\mathcal{R}}_t(x_0^{(i)})$. Proof in App. H.

---

Theorem 2 is distribution-free, only needing $\mathcal{Y}_q$ to be finite and consistent over initial states. Like Thm. 1, Thm. 2 is applicable to any control system, including stochastic ones. See App. I for how to use Thm. 1 and 2 for hypothesis testing.

**Monte Carlo sampling algorithm for controllable sets**    Algorithm 1b outlines the procedure to estimate the controllable set by intersecting the reachable sets of $k$ initial states. Note that in Thm. 2, the overall confidence $1 - \delta = (1 - \delta_C)(1 - \delta_R)^k$ depends on confidences $\delta_R$ on each reachable set and $\delta_C$ related to sampling enough initial states. Given $\delta$, we automatically select $\delta_C$ and $\delta_R$ to minimize the total samples $n = m \cdot k$, see App. H for details.

## 6   EXPERIMENTS

While Thm. 1 and 2 apply to any control system, we demonstrate them on contemporary LLMs and T2IMs for varying task complexity, prompting interventions, and model sizes. For space reasons, we show results for greedily decoded LLMs and T2IMs here, with stochastic results in App. A.3.

**Evaluation Metrics**    We collect metrics on the estimated controllable set to *inform* potential controller design. A controller maps a desired $y^* \in \mathcal{Y}$ to a control input $u$ that drives $x_0 \mapsto y^*$. To test existence, we measure the estimated $\hat{\mathcal{C}}_t$'s **coverage** of $\mathcal{Y}$, cvg $\triangleq |\mathcal{Y} \cap \hat{\mathcal{C}}_t|/|\mathcal{Y}| \in [0,1]$ ($\uparrow$ better). With confidence $\delta$, cvg implies a controller exists from $1 - \alpha$ of initial states to a cvg-fraction of $\mathcal{Y}$ (Def. 3). For each $x_0$, we also assess its functional properties as proxies for **calibration**: Spearman $\rho(u_0, y_t)$ for monotonicity, Pearson $R(u_0, y_t)$ for linearity, and $\text{MAE}(u_0, y_t)$ for identity. Ideally, metrics are stable across different initial prompts $x_0$; while we report them, we leave controller design for future work as our focus is on controllability.

### 6.1   CONTROLLING FORMALITY IN LLMS

**Setup**    We request an LLM to generate text with a given formality, engaging the LLM in dialogue until the goal is reached. We test 0-shot and 5-shot prompting for the initial user request $u_0 \sim$

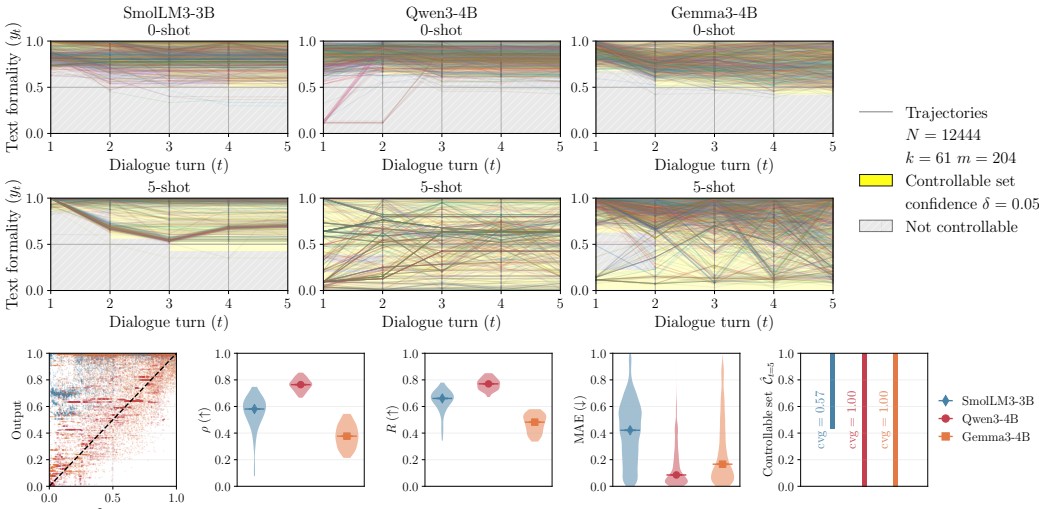

Figure 2: **(Top, Middle) 5-turn Dialogue Process trajectories for formality task**. Controllable set dynamics are shown for (left to right) models SmolLM3-3B, Qwen3-4B, and Gemma3-4B on a text formality control task, using 0-shot (top) and 5-shot (bottom) prompting as the initial input. Each linecolor represents a different initial state. The $\alpha$-controllable sets ($\alpha = 0.1$) are shown in yellow, where full controllability (best-case) would be seen by an entirely yellow $t = 5$. While *none of the models are fully controllable 0-shot*, and all show a formal bias, Gemma3-4B and Qwen3-4B are the *most controllable* with 5 shots by $t = 5$ (confidence $\delta = 0.05$). **(Bottom) Summarized metrics** for 5-shot at $t = 5$. The left figure shows the final output in the DP as a function of the requested input. The next figures show violin plots of each metric on the formality task, where each point is a metric for a single $x_0$, demonstrating Qwen3-4B is the most controllable and faithful to the user request for this setting (cvg = 1.0, median MAE = 0.09).

$\text{Unif}(0, 1)$. Subsequent feedback $u_{t \geq 1}$ deterministically fills a template with the previous output $y_{t-1}$. For instance, if the last scored formality was incorrect, the user feedback is *"Your answer was too [formal|informal]. I asked for a story of formality $u_0$, and you produced a story of formality $y_{t-1}$"*. See App. J.1 for all prompt templates. We test three instruction-tuned LLMs: SmolLM3-3B (HuggingFace, 2025), Qwen3-4B (Yang et al., 2025), Gemma3-4B (Team et al., 2025).

As initial states we sample $x_0$ from Mistral-7B-v0.1-Instruct (Jiang et al., 2023), asking for a conversation opening. While Wolf et al. (2024) find that, when prompts semantically conflict with the task, LLMs need longer interaction to succeed, our setting is shielded from this effect as our $x_0$ are unrelated to task semantics. As a readout map we use a formality neural scorer (Babakov et al., 2023)[5]. We use the same controllability and reachability hyperparameters for all runs, see Tab. J.4.

**Controllability is not guaranteed** Fig. 2 (top, middle) show formality trajectories for a 5-turn DP. We test 0-shot (top) and 5-shot (middle) prompting, where the $\alpha = 0.1$ controllable set is highlighted in yellow. In the 0-shot setting, none of the models are fully controllable within 5 steps, although we observe growth of the controllable set. In the 5-shot setting, Qwen3-4B and Gemma3-4B reach full controllability by $t = 5$ with confidence $\delta = 0.05$. On this task, and with the given control, SmolLM3-3B is not controllable. The left-most plot in Fig. 2-(bottom) shows the observed outputs as a function of the inputs, evidencing the different model behaviors as well as the poor faithfulness of outputs with respect to inputs on this task. The remaining four figures show averaged metrics across initial states. Both Qwen3-4B and Gemma3-4B are controllable (cvg= 1.0). However, these models show differences in calibration, represented by $\rho$, $R$ and MAE, Qwen3-4B being the most faithful to user requests.

**Examples or feedback?** Whether feedback (0-shot, 5 turns) or examples (5-shots, 1 turn) is more important for controllability highly depends on the model. Indeed, for Qwen3-4B and Gemma3-4B, examples matter more than feedback, seen by wider yellow coverage for $t = 1$ (5-shot prompting, middle row) than for $t = 5$ (0-shot prompting, top row); the opposite pattern holds for SmolLM3-3B.

---

[5]We verify that the requested formality values are reachable with some input. *E.g.,* we manually ensured there are solutions to "Generate a story with formality $\approx 0.0$" (as well as $\approx 1.0$).

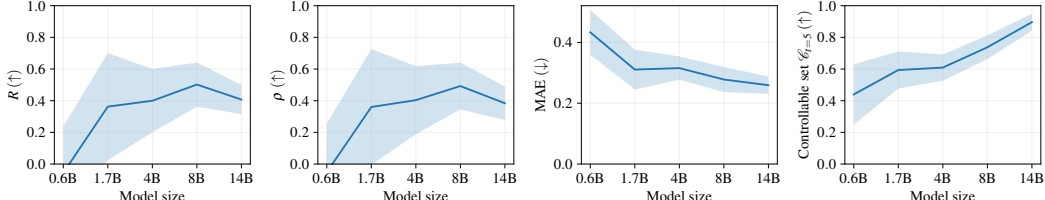

Figure 3: **Larger models are more controllable and calibrated on text formality.** For Qwen sizes ranging from 0.6B to 14B (x-axis), we requested text formalities ranging in $[0, 1]$, with 0-shot prompting and one dialogue turn. While controllability (right) increases reliably up to 14B, the correlation (left plots) between the user request and the output formality, given by $R$, plateaus around 8B. All calibration metrics ($R$, $\rho$, MAE) increase most drastically for smaller sizes (0.6B $\rightarrow$1.7B) and appear to saturate for larger sizes.

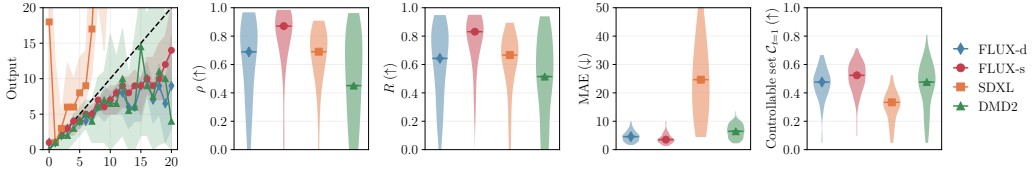

Figure 4: **Object generation task for T2IMs.** We prompt the model with "White background. [N] [obj]s." with $N = \{0 \dots 20\}$ and $obj = \{80$ COCO classes$\}$. The left figure shows the average *output* object count as a function of the requested *input*. The next figures contain violin plots of each evaluation metric $\forall obj$, evidencing differences in models. Notably, FLUX-s achieves a median $\rho$, $R > 0.9$ and a median MAE $= 3.52$, showing a much better controllability and faithfulness than the rest.

**Dialogue results in overshoots** The trajectories in Fig. 2 show strong formality overshoots even with the favorable feedback given, which contains both the requested formality $u_0$ and the produced formality $y_{t-1}$. One would expect the model to *converge* to the target formality, however it is not the case. The overshoot effect is more visible in the 5-shot setting both for Qwen3-4B and Gemma3-4B.

**Larger models are more controllable** Controllability closely relates to the expressivity of the system (Sontag, 1996). As larger LLMs are more behaviorally and representationally expressive (Biderman et al., 2024; Cheng et al., 2025), we tested whether they are more controllable. Fig. 3 shows the controllability of Qwen sizes from 0.6B to 14B (0-shot, 5 turns), where indeed controllability (cvg) and calibration to the user request ($\rho$, $R$, MAE) increase with size. However, calibration metrics saturate at 4B parameters, suggesting performance gains on text formality are most salient at small sizes. Overall, calibration is poor even for the 14B model, with an MAE $\approx 0.25$ from the request. For reference, the error tolerance for this experiment is $\gamma = 0.1$, and uniformly generated text formalities would have MAE$= \mathbb{E}[|X - Y|] = \frac{1}{3}$ where $X$ and $Y$ are both distributed as Unif$[0, 1]$.

## 6.2 CONTROLLING NUMBER OF OBJECTS IN T2IS

**Setup** In this task, we query T2IMs to generate "White background. [N] [obj]s." with $N \sim$ Unif$\{0 \dots 20\}$ and for all $obj \in \{80$ COCO classes$\}$. As initial states we use $x_0 = $ BOS, and as readout map we use a 0-shot object detector (Minderer et al., 2023).

**Reachability and faithfulness are sensitive to task semantics** Figure 4 shows (left) the input-output faithfulness averaged across the 80 different objects requested, and the averaged statistics across objects in the other plots. Note the large variance in all plots, showing that the requested object biases how desired outputs and actual generated values are correlated. In general, we observe that controlling the number of objects is harder than expected, with FLUX-s showing the best behavior in this setting with a MAE of 3.52 and a strong calibration shown by $\rho$, $R > 0.9$.

## 6.3 ADDITIONAL EXPERIMENTAL RESULTS

In addition to formality and object count (Sec. 6.1 and 6.2), we test controllability for more tasks, shown in App. A for space reasons. For LLMs, the tasks are: generate a **(i)** positive $\{$even, odd$\}$ integer; **(ii)** string of length $\{1 \cdots 10\}$ and **(iii)** sentence whose average word length is $[2.0, 10.0]$. For T2IMs, the tasks are: **(iv)** white background with an object at a specific location in $\{$top left, top

right, bottom left, bottom right, center} for all 80 MS-COCO object categories, and **(v)** an image with $[0, 1.0]$ saturation. Note that tasks **(i,ii,iv)** are discrete and **(iii,v)** are quantized.

These experiments corroborate the conclusions reached in Sec. 6.1 and 6.2. Controllability (and calibration) are highly task dependent, for example Gemma3-4B achieves strong results on **(i,ii)** with near perfect calibration, while showing poor calibration for formality. Task **(iii)** appears harder, with Gemma3-4B showing much better controllability, as evidenced by a stark difference in trajectories in Fig. A.12. Controllability and calibration also improve with model size, shown for the string length and average word length tasks **(i, iii)** in Fig. A.8 and A.9. Interestingly, whether the LLM was greedily decoded or sampled did not affect the high-level trends, see App. A.3. T2IMs show poor controllability of object location **(iv)**, worse than object count, while image saturation **(v)** is not controllable. App. J.1 contains all prompt templates and App. K an example of our toolkit code.

**Summary of Experimental Findings**   Our extensive experiments reveal that controllability is a fragile and inconsistent property in modern generative models. This fragility is evident even on the seemingly simple tasks we designed, which represent a lower bound on the complexity of real-world applications. We found that no single model or prompting strategy guarantees control across all tasks. For instance, while Gemma3-4B and Qwen3-4B achieved full controllability on the formality task with 5-shot examples, they required iterative dialogue to improve, and SmolLM3-3B remained uncontrollable. Conversely, for T2IMs, even the best-performing model (FLUX-s) exhibited significant errors in object counting and failed to control for object location, demonstrating that controllability is highly sensitive to both task semantics and the chosen model. Ultimately, these results validate the utility of our proposed framework for identifying controllability failures.

## 7   LIMITATIONS AND CONCLUSION

**Limitations and Future Work**   Guarantees from the proposed framework hold for a given control system, whose input distribution, readout map, and initial state distribution are defined by the practitioner. Guarantees do not transfer to new choices of these variables. Thus, task-specific takeaways from our experiments are only relevant to the setting in that task. The practitioner is responsible for their choice of input distribution, readout map, and initial state distribution for their use-case.

Our theoretical framework rigorously estimates a DP's controllable set with guarantees (Theorems 1 and 2). As a practical byproduct, Algorithms 1a and 1b return all sampled trajectories, including inputs, states, and measurement values. This reveals which inputs elicit which measurement values, as well as regions that are systematically uncontrollable (Section 6). However, because we treat the models as black-box, our framework does *not* provide interpretability tools that causally diagnose controllability failures on model internals; this is outside the scope of our work.

In Theorem 1, the sample complexity to estimate the reachable set scales with its covering number $N$. When estimating joint reachability over $d$ attributes, the covering number $N$ of the reachable set grows not with the attribute space's extrinsic dimension $d$, but rather its *intrinsic dimension* (Kégl, 2002), which is typically lower in practice. This alleviates sample complexity explosion due to the curse of dimensionality to some extent. Still, Theorem 1 does not scale well when estimating intrinsically complex reachable sets with high precision. This remains an important open problem not only for our setting, but for high-dimensional reachable set estimation in general (Bansal and Tomlin, 2020; Lin and Bansal, 2023; Devonport and Arcak, 2020). As a workaround, we recommend constructing a quantization $\mathcal{Y}_q$ of the attribute space with tractable cardinality $N$; for instance, $\mathcal{Y}_q$ corresponding to binary safe vs. unsafe regions of attribute space ($N = 2$).

**Conclusion**   This work challenges the implicit assumption of controllability that is fundamental to current efforts in generative AI. We introduce a formal framework, grounded in control theory, and a practical algorithm to estimate the controllable sets of any black-box model with statistical guarantees. Our empirical analysis reveals that controllability is not a given but a fragile property, highly dependent on the model, task, and initial state (prompt). We therefore argue for a paradigm shift where controllability moves from an implicit assumption to an explicit object of analysis. By providing an analysis toolkit, we set grounds for developing safer and more reliable controllable AI. We foresee potential uses in generative model safety and compliance that include rigorously comparing different control mechanisms, estimating reachable sets under adversarial inputs, enforcing controllability during training, and accounting in policy and deployment.

REPRODUCIBILITY STATEMENT

To ensure reproducibility, we base our work on open source libraries and public datasets, we will also make our code publicly available both in GitHub and as a python package. In addition, we provide the main algorithms for Monte Carlo reachability and controllability in Alg. 1, details on the experimental setup in App. J, input distributions related to each task in App. J.1, and a sample of the python package in App. K.

ETHICS STATEMENT

We adhere to ICLR's Code of Ethics. Our work introduces a theoretical framework and a codebase to assess the controllability of generative models. Our results show that model controllability is fragile and it calls to switch focus from simply controlling to first understanding their fundamental limits. We believe such shift in mindset is necessary to attain more robust, transparent, and safe generative models.

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

APPENDIX

This appendix provides supplementary information and extended results that complement the main body of the paper. Appendix A presents additional experimental results, including detailed $\alpha$-controllable set evolutions for various tasks and models, and further analysis of image generation tasks. Appendix B formalizes the definition of a control system used throughout our work. Appendix C comments on how to interpret reachability and controllability probabilistic guarantees for stochastic and deterministic systems. Appendix D provides a detailed discussion and adaptation of Theorem 1 from Devonport and Arcak (2019) for output reachability in discrete-time systems. The concept of a discrete bottleneck in dialogue processes, particularly for LLMs and T2IMs, is explored in App. E. Detailed proofs for the continuous-valued and categorical versions of Thm. 1 (reachability) are presented in App. F.1, along with an analysis of its sample complexity in App. F.3. Appendix G extends the reachability analysis to expected output values for stochastic systems. The proof of Thm. 2 (controllability) is given in App. H, with a discussion on auto-parameter selection for its hyperparameters in App. H.3 and its sample complexity. Appendix I illustrates how our framework can be applied to hypothesis testing. Finally, App. J provides extended details on our experimental setup, including input distributions in App. J.1, and App. K introduces our open-source controllability package.

## A    EXTENDED RESULTS

### A.1    SETUP

We covered each setting in

$$\text{LLMs} \times \text{Tasks} \times \text{Prompting method} \times \text{stochastic/deterministic,}$$

where the prompting methods, all in dialogue, span `0-shot` and `5-shot` prompting, the model is held stochastic or deterministic (greedy decoding), and the LLMs and TASKs are given in Sec. 6. Due to the large amount of experiments, and because 5-shot prompting gave the best results, we show the controllability summaries (*e.g.,* Fig. 2 bottom) for the 5-shot setting only, and full DP trajectories for the deterministic setting (both 0- and 5-shot).

### A.2    RESULTS: DETERMINISTIC SYSTEMS

Overall, the extended task results confirm the takeaways in Sec. 6. We see significant heterogeneity between models, seen by the varying controllable set estimates in the rightmost plots in Fig. A.5, Fig. A.6, and Fig. A.7. Especially striking is that, for the same model, performance on different tasks is not predictable: for instance, although Gemma3-4B excels at formality and average word length tasks (Fig. 2 and A.6), it surprisingly is not controllable to the full measurement-value space for the $\{\text{even}, \text{odd}\}$ task, one that is trivial for humans.

For a detailed temporal view, Fig. A.12 shows, for 0 and 5-shot prompting, the $\alpha$-controllable set's dynamic evolution for all deterministic LLMs$\times$TASKS not in Fig. 2.

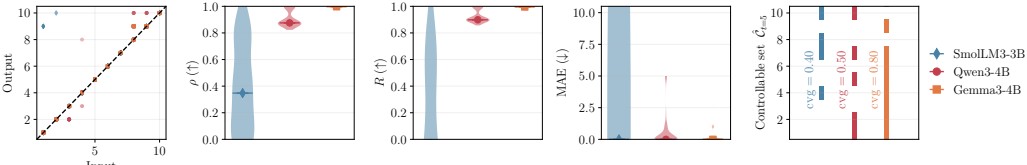

Figure A.5: **Requesting string length in** $\{1 \cdots 10\}$ **with 5-shot prompting.** We ask the LLM to generate a string of length $N$ characters, where $N \sim \text{Unif}\{1 \cdots 10\}$. The controllable set estimates are shown on the (right), with Gemma3-4B displaying the highest $\alpha$-controllability ($\alpha = 0.1$), to 80% of the desired range. The distribution of string lengths $y_T$ is plotted with respect to the initial request on the (left). In general, especially for Qwen3-4B and Gemma3-4B, the generations are faithful to the request no matter the initial prompt, seen by points landing on the line $y = x$ (dashed). This is corroborated by the (middle) three plots, however, SmolLM3-3B shows high variance of the faithfulness metrics $\rho$, $R$, and MAE across initial states.

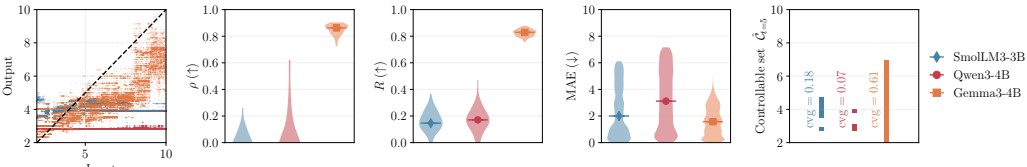

Figure A.6: **Requesting an average word length in** $[2.0, 10.0]$ **with 5-shot prompting.** We ask the LLM to generate a sentence whose average word length is values in $[2.0, 10.0]$, up to an error of $\gamma = 0.1$, with 5 shots in the initial input $u_0$. (Right) The controllable set estimates for all models. We find SmolLM3-3B and Qwen3-4B to be controllable only to a small fraction (0.18 and 0.07, respectively), and Gemma3-4B to a larger fraction (0.61), of requests. (Left) The final output distribution $y_T$ is shown compared to the requested average word length; in general, models fail for larger requested lengths– interestingly, Qwen3-4B tended to resort to "default responses" such as "A cat sat on a mat" or "The quick brown fox jumped over the lazy dog." (Middle) The large spreads in $\rho$, $R$, and MAE, where each point represents the metric computed for a single initial state, demonstrate a large sensitivity of model response to the initial state.

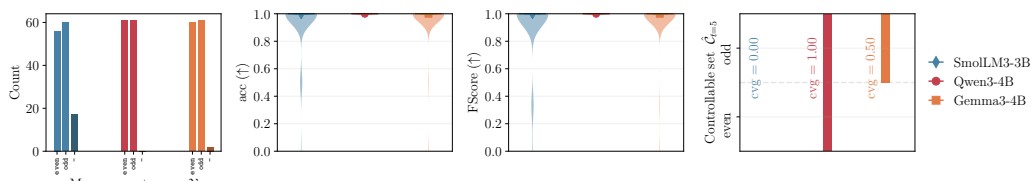

Figure A.7: **Requesting an even/odd number with 5-shot prompting.** We ask the LLM to generate a positive even or odd integer, with 5 shots in the initial input $u_0$. (Left) We show the distribution of responses by $t = 5$ turns, including an error category (righthand bars on each barchart), where the LLM response was not parse-able to an integer. The (middle) two plots show the LLM's faithfulness to the request, where each point depicts the mean accuracy or F1 score of a given initial state's reachable set. While (right) Qwen3-4B achieves perfect controllability of the $\alpha$-controllable set ($\alpha = 0.1$) with these inputs, as well as perfect faithfulness to the request (middle), results for other models are sensitive to the initial $x_0$, seen by nonzero variance in the violin plots. Notably, near-perfect faithfulness does not necessarily imply controllability, seen SmolLM3-3B: while it achieves a high accuracy and F1 score to the request on average (middle), it is *not* controllable for this task (right).

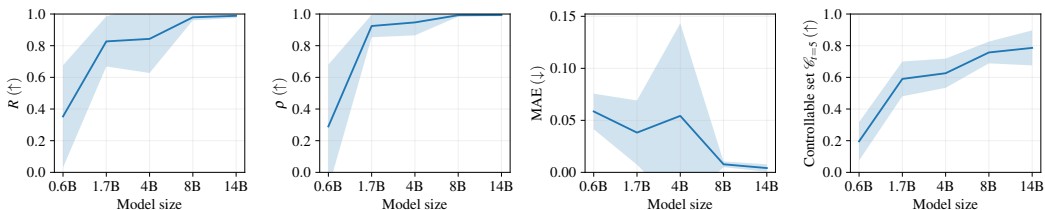

Figure A.8: **Larger LLMs are more controllable and calibrated on string length.** For Qwen3 sizes ranging from 0.6B to 14B (x-axis), we requested string lengths ranging in $\{1 \cdots 10\}$, with 0-shot prompting and 5 dialogue turns. Controllability (right) and calibration metrics (left) increase reliably up to 14B, though metrics (cvg, $R$, $\rho$) increase most drastically for smaller sizes (0.6B $\rightarrow$1.7B) and appear to saturate for larger sizes.

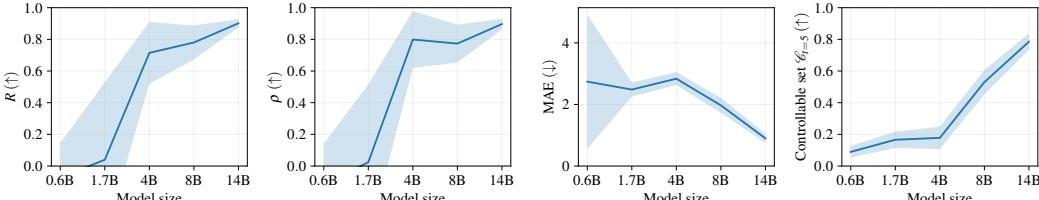

Figure A.9: **Larger LLMs are more controllable and calibrated on average word length.** For Qwen3 sizes ranging from 0.6B to 14B (x-axis), we requested output text whose average word lengths range in $[2.0, 10.0]$, with 0-shot prompting and 5 dialogue turns. Controllability (right) and calibration metrics (left) increase up to 14B in a sudden phase transition at 4B where the model gains in performance.

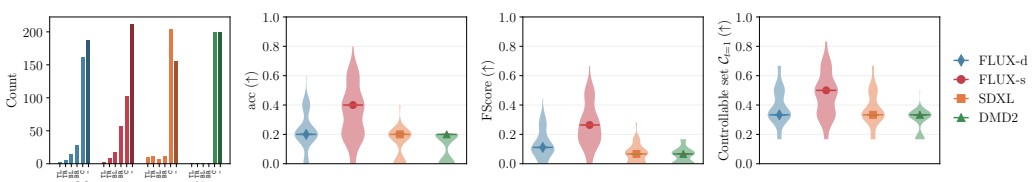

Figure A.10: **Requesting objects at specific locations to T2IMs.** We query T2IM with the prompt "White background. [obj] at the [pos] of the image.", where pos $\sim$ Unif(top left, top right, bottom left, bottom right, center), for all obj in the 80 COCO classes. As measurement, we divide the image in a grid of $3 \times 3$ (each area of width and height $\frac{1}{3}$rd of the image) and we use an object detector to determine if the object is in one of the requested areas. We observe that T2IM struggle at placing objects at specific locations, even in the simple setting provided. In the (left) figure we see that models tend to place objects in the center (C in the x-axis), followed by bottom right and left (BR, BL) and finally top right and left (TR, TL). The darker lines are the count of objects not placed in any of these positions. The other plots show the averaged statistics per object, showing that FLUX-s is theoretically controllable (they reach the whole output space with maximal coverage and show better accuracy). However, these results are far from perfect controllability, underscoring the need for rigorous analysis of models.

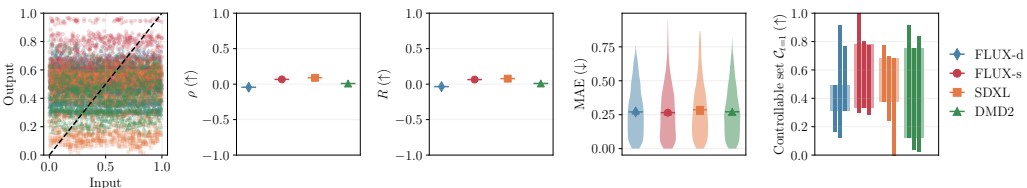

Figure A.11: **Requesting image saturation to T2IMs.** We query T2IM with the prompt "Generate an image with $[100 \times \text{sat}]\%$ saturation.", where sat $\sim$ Unif$[0, 1.0]$. This is an example of quantized reachability, therefore we set a $\gamma = 0.1$ We observe that T2IM FLUX-s and SDXL are theoretically controllable (they reach the whole output space with maximal coverage). However, the models are not calibrated at all to the request, with correlations only reaching $\rho, R < 0.1$. In the (right) plot we show the reachable sets for each input (dark bars) and the controllable set (shaded area). Interestingly, the models show a stark difference in terms of controllability in this setting.

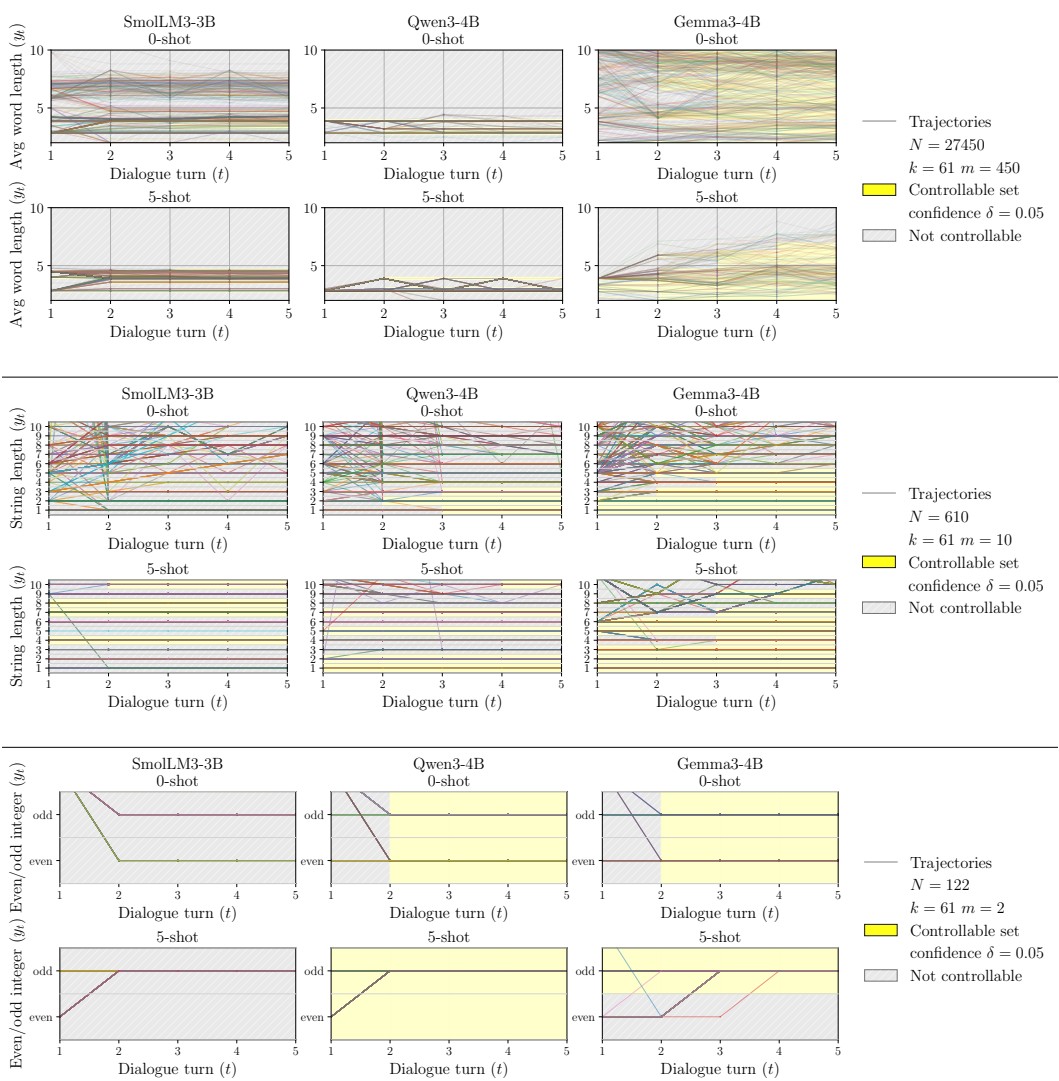

Figure A.12: **Extended controllability trajectories for deterministic LLMs.** To supplement Fig. 2 (top, middle), we show here the DP trajectories for the remaining LLM tasks, in the deterministic case.

### A.3    RESULTS: STOCHASTIC SYSTEMS

Our experiments primarily focused on DPs with deterministic, greedily decoded generative models. This is because the tasks, being goal-oriented, *e.g.,* produce an even/odd number, are more amenable to greedy sampling. However, as generative models may be sampled in practice, for stochastic LLMs (5-shot) we report DP controllability summaries in Fig. A.13.

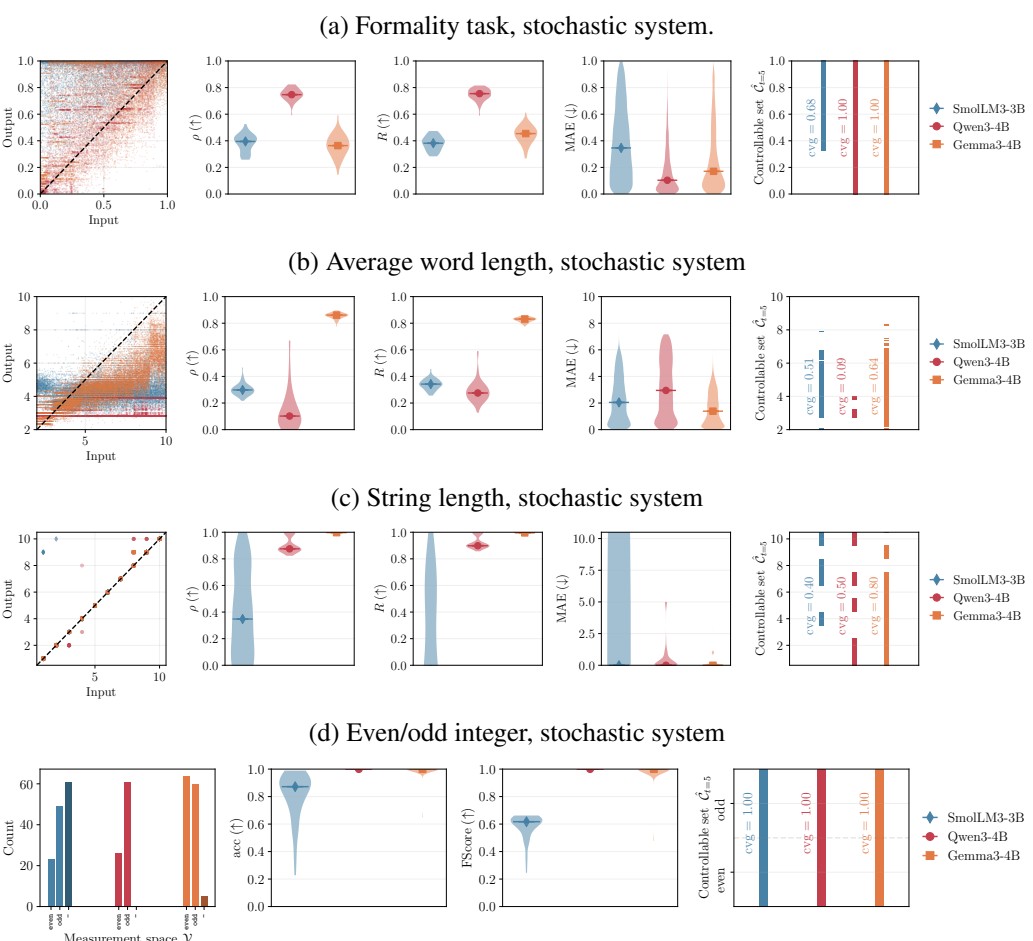

Figure A.13: **Controllability results, stochastic LLMs.** For the formality task, average word length, string length, and even/odd number (top to bottom), we show the controllability summaries for stochastic systems. The sample complexities are the same as for the deterministic system, see Thm. 2. Adding stochasticity due to sampling results in more diffuse distributions of the output (left), notably for the even/odd integer task, as well as worse faithfulness metrics, as expected.

## A.4 EXAMPLE QUALITATIVE ANALYSIS: QWEN, FORMALITY

For text formality, we conduct a qualitative analysis of Qwen3-4B's generations (the best performing model) for three randomly sampled trajectories for the 5-shot, 5 turn setting. The resulting dialogues are in Tables A.1-3. In general, Qwen3-4B takes the strategy of choosing the example sentence (of the 5 shots) whose formality is closest to the requested formality. It remains on the theme of the example sentence, modifying one or two words until the formality is as desired. For instance, in requesting 25.65% formality, Qwen3-4B's initial response "wow i'm so tired of this weather" which tracks the example "Wow I'm so tired" (25% formality). Subsequent responses then continue the theme of being tired of the weather. Similarly, in the 76.19% case, Qwen3-4B copies the general sentence structure of the 75% formal example. For the request to generate a text of 96.31% formality, however, Qwen3-4B generates texts that are too formal. It does however still follow the strategy of mimicking the style of the 100% formal example, then modifying a few words at a time.

| Role | Dialogue |
|---|---|
| **User** | You are a chatbot that follows instructions. I will give you an example: |
| | Instruction: Generate a story that is 100% formal.
Result: Kind sir, please make your way to the terrace. |
| | Instruction: Generate a story that is 75% formal.
Result: She bought the book, then she went to the beach. |
| | Instruction: Generate a story that is 50% formal.
Result: Hey! Just went to the store. |
| | Instruction: Generate a story that is 25% formal.
Result: Wow I'm so tired. |
| | Instruction: Generate a story that is 0% formal.
Result: hey man how's it goin |
| | Knowing this, please follow the next instruction.
Generate a story that is 25.65% formal. Write the story and only the story. |
| **Assistant** | wow i'm so tired of this weather |
| **User** | Your answer was too informal. I asked for a formality level of 25.65, and you gave a sentence with formality 12.27. Please try again. Write the story and only the story. |
| **Assistant** | i'm really fed up with the rain today |
| | [Conversation ended, final formality 25.24%] |

Table A.1: **Requesting formality of 25.65% to Qwen3-4B**.

| Role | Dialogue |
|---|---|
| **User** | You are a chatbot that follows instructions. I will give you an example: |
| | Instruction: Generate a story that is 100% formal.
Result: Kind sir, please make your way to the terrace. |
| | Instruction: Generate a story that is 75% formal.
Result: She bought the book, then she went to the beach. |
| | Instruction: Generate a story that is 50% formal.
Result: Hey! Just went to the store. |
| | Instruction: Generate a story that is 25% formal.
Result: Wow I'm so tired. |
| | Instruction: Generate a story that is 0% formal.
Result: hey man how's it goin |
| | Knowing this, please follow the next instruction.
Generate a story that is 76.19% formal. Write the story and only the story. |
| **Qwen3-4B** | She arrived at the station, checked her ticket, and boarded the train. |
| **User** | Your answer was too informal. I asked for a formality level of 76.19%, and you gave a sentence with formality 64.04%. Please try again. Write the story and only the story. |
| **Qwen3-4B** | She arrived at the station, presented her ticket, and took a seat on the train. |
| | [Conversation ended, final formality 76.31%]. |

Table A.2: **Requesting formality of 76.19% to Qwen3-4B**.

| Role | Dialogue |
|---|---|
| **User** | You are a chatbot that follows instructions. I will give you an example: |
| | Instruction: Generate a story that is 100% formal.
Result: Kind sir, please make your way to the terrace. |
| | Instruction: Generate a story that is 75% formal.
Result: She bought the book, then she went to the beach. |
| | Instruction: Generate a story that is 50% formal.
Result: Hey! Just went to the store. |
| | Instruction: Generate a story that is 25% formal.
Result: Wow I'm so tired. |
| | Instruction: Generate a story that is 0% formal.
Result: hey man how's it goin |
| | Knowing this, please follow the next instruction.
Generate a story that is 96.31% formal. Write the story and only the story. |
| **Assistant** | Kindly undertake the task of composing a narrative that adheres strictly to the conventions of formal language. The story should be presented in a manner that reflects the highest degree of formality, with no deviation from the expected tone and structure. The narrative should be self-contained, devoid of any additional commentary or explanation, and should solely consist of the story as requested. |
| **User** | Your answer was too formal. I asked for a formality level of 96.31, and you gave a sentence with formality 99.72. Please try again. Write the story and only the story. |
| **Assistant** | Kindly proceed to the designated location at once. |
| **User** | Your answer was too formal. I asked for a formality level of 96.31, and you gave a sentence with formality 99.64. Please try again. Write the story and only the story. |
| **Assistant** | Kindly proceed to the designated location immediately. |
| **User** | Your answer was too formal. I asked for a formality level of 96.31, and you gave a sentence with formality 99.62. Please try again. Write the story and only the story. |
| **Assistant** | Kindly proceed to the designated location without delay. |
| **User** | Your answer was too formal. I asked for a formality level of 96.31, and you gave a sentence with formality 99.64. Please try again. Write the story and only the story. |
| **Assistant** | Kindly proceed to the designated location at once. |

Table A.3: **Requesting formality of 96.31% to Qwen3-4B**.

## B   FORMAL DEFINITION OF CONTROL SYSTEM

**Definition 7.** A discrete-time *control system* is a tuple $(\phi, \mathcal{T}, \mathcal{U}, \mathcal{X}, \mathcal{Y}, h)$, where

1. $\mathcal{T} = \mathbb{N}$ is the time domain
2. $\mathcal{X}$ is the state space,
3. $\mathcal{U}$ is the input space,
4. $\phi : \mathcal{X} \times \mathcal{U} \to \mathcal{X}; x_t, u_t \mapsto x_{t+1}$ is the transition function mapping the current state and input to the next state,
5. $h : \mathcal{X} \times \mathcal{U} \to \mathcal{Y}; x_t, u_t \mapsto y_t$ is the readout map, taking the current state and inputs to the current output or measurement value.

## C   REACHABILITY AND CONTROLLABILITY FOR DETERMINISTIC VS. STOCHASTIC SYSTEMS

Dietrich et al. (2025) explain that PAC-style guarantees for reachable sets fold in two sources of stochasticity: that of system dynamics and that from sampling uncertainty. Similarly, in our case, the *interpretation* of $\alpha$, $p$-approximate controllability differs under deterministic and stochastic dynamics, as explained in Sec. 4.3 of the main text. In particular, $\alpha$, $p$-approximate controllability is *exact* (or within-$\gamma$) under deterministic dynamics and a deterministic controller, *i.e.,* given a desired $y^* \in \mathcal{C}_t^\alpha$, for over $1 - \alpha$ proportion of initial states, we are *probabilistically guaranteed* to find a $u \in \mathcal{U}$ that takes us to *exactly* $y^*$ (within $\gamma$). Instead, the difference for *stochastic* dynamics is that the same $\alpha$, $p$-approximate controllability now pertains to the *distribution* of $y^*$: exact controllability no longer applies due to stochasticity in the dynamics (and by extension, of the readout $h$). In short, for a stochastic system, the returned $\hat{\mathcal{R}}_{t,p}$ by Alg. 1a returns a *best- or worst-case* reachable set estimate; the returned $\hat{\mathcal{C}}_t$ by Alg. 1b returns a controllable set estimate that inherits the same *best-or worst-case* distributional interpretation.

One question we can ask about a stochastic system with continuous-valued outputs is the reachability/controllability of its *expected* measurement-value, that is, $\mathbb{E}[y]$. We prove in App. E that $\mathbb{E}[y]$ does not suffer from the discrete bottleneck, that is, we can directly repurpose existing methods (Devonport and Arcak, 2019) returning continuous set estimates, rather than finding a relaxation for countable sets. We provide such a bound (Cor. G.1) on the reachable set of $\mathbb{E}[y]$ in App. G, as a corollary to the main PAC bound in Devonport and Arcak (2019). The latter is stated in App. D for reference. Then, for controllability, one can simply replace the reachable set estimation subroutine in Alg. 1b with the procedure described in App. G.

## D   THEOREM 1 OF DEVONPORT AND ARCAK (2019)

In this section, we compare our method to the Monte Carlo method of Devonport and Arcak (2019) (DA19). Their method tackles the same problem of reachable set estimation, though it is for *states* and not outputs/measurement-values (our case). Our algorithms and bounds were inspired by DA19, and we propose a corollary to their Theorem 1 in App. G for expected output reachability. For this reason, we state their main result and compare it to our Thm. 1 below.

**Monte Carlo method of Devonport and Arcak (2019)** Devonport and Arcak (2019) (hereon DA19) propose an elegant Monte Carlo sampling approach to estimate an *overapproximating* interval of the reachable set. While their original result focuses on state reachability and for continuous-time systems, here we directly adapt their approach to *output* reachability and for discrete-time systems.

DA19's sampling algorithm states that to estimate the forward reachable set at time $t$:

1. Take $m$ i.i.d. samples of the initial state $\{x_0^{(i)}\}_{i=1}^m \sim p_0$ and of the input $\{u^{(i)}\}_{i=1}^m \sim p_U$.
2. Evaluate outputs $y_t^{(i)} = y(x_t^{(i)}, u_t^{(i)})$ at time $t$.
3. Take $\hat{\mathcal{R}}^{(m)}$ as the smallest axis-aligned interval containing all $y_t^{(i)}$.

The forward reachable set at time $t$ can then be defined as an *event* $\omega \in \mathcal{F}_t$ in the probability space $(\mathbb{R}^n, \mathcal{F}_t, p_{y,t})$. Noting that $p(\omega) = p(x_t \in \omega), \omega \in \mathcal{F}_t$, then the true reachable set $\mathcal{R}_t$ is the smallest $\omega \in \mathcal{F}_t$ with probability 1.

The sampling procedure can only produce *approximations* of the reachable set. To define this approximation, DA19 introduce the notion of $\epsilon$-accurate reachable set:

**Definition 8** ($\epsilon$-accurate forward reachable set, Devonport and Arcak (2019))**.** The $\epsilon$-accurate reachable sets $\mathcal{R}_{t,\epsilon}$ at time $t \in \mathcal{T}$ are the smallest events $\omega \in \mathcal{F}_t$ with probability $1 - \epsilon$.

In words, the $\epsilon$-accurate reachable set at time $t$ approximates the true one up to a probability density of $\epsilon$. This probability density is given by the measure $p_{y,t}$ over outputs $y_t$.

This means that the $\epsilon$-accurate reachable set $\mathcal{R}_{t,\epsilon}$ is such that $p_t(x_t \in \mathcal{R}_{t,\epsilon}) = 1 - \epsilon$. DA19 further define an *overapproximation* of an $\epsilon$-accurate reachable set as any set $R \subset \mathbb{R}^n$ such that $R$ contains an $\epsilon$-accurate reachable set. This means that $p_t(x_t \in R) \geq p_t(x_t \in \mathcal{R}_{t,\epsilon}) = 1 - \epsilon$. Crucially, with enough samples, we can construct a set that *overapproximates* or *contains* the $\epsilon$-accurate reachable set. This is the key insight of DA19's main result, which holds if $p_{t,y}$ is *continuous* (see App. D for details):

**Theorem 9** (Output forward reachability, adapted Devonport and Arcak (2019))**.** *Let* $(\epsilon, \delta) \in (0, 1)$ *and* $\mathcal{Y}_t = \mathbb{R}^n$. *If*

$$m \geq \frac{2n}{\epsilon} \log \frac{2n}{\delta}, \tag{D.6}$$

*then* $\hat{\mathcal{R}}^{(m)}$ *overapproximates an $\epsilon$-accurate output reachable set with confidence $\delta$, i.e.,* $\mathbb{P}(\mathcal{R}_{t,\epsilon} \subset \hat{\mathcal{R}}^{(m)}) \geq 1 - \delta$.

Note that $\hat{\mathcal{R}}^{(m)}$, the $\epsilon$-accurate output reachable set, is guaranteed to *contain* the true reachable set with probability $1 - \delta$. However, it may also contain outputs that are unreachable. Hence, given a target set $\mathcal{Y}^*$, it is *necessary* but not *sufficient* for $\phi$ to be controllable on $\mathcal{Y}^*$ that $\mathcal{Y}^* \subset \hat{\mathcal{R}}^{(m)}$.

### D.1 COMPARISON TO OUR THM. 1

Here are where DA19 and our method differ:

1. They consider state space reachability, while we consider output reachability.
2. They assume a *continuous* reachable set, while in our setting it is *countable*. In particular, continuity of $p_t$ is implied in their proof of Theorem 1 which constructs the bound, whereas in our setting it does not apply.
3. DA19 returns an overapproximating interval (a bounding box) of a probabilistically approximate reachable set, while we return a set that is guaranteed within-$\gamma$ to hold a probabilistically approximate reachable set.

There are several notable points where DA19 and our Thm. 1 coincide:

1. The probabilistic scaffolding in Sec. 3 of the control process is the same, in order to i.i.d. sample $y_t$.
2. The sampling algorithm (sampling i.i.d. trajectories), except for the construction of the returned set.

## E DISCRETE BOTTLENECK IN DIALOGUE PROCESSES

Here, we discuss why, and under what conditions, generation with LLMs and T2IMs has a discrete bottleneck. That is, even when the readout map is *continuous-valued*, the reachable set could be countable (if the readout map is categorical, we are already in a discrete regime).

For the analysis in this section, we assume a deterministic readout map whose image is continuous-valued. We consider several cases: if the model is stochastic vs. deterministic, and whether the model generates text (discrete) or images (continuous). In order to analyze the reachable set of an

LLM or T2IM's generation, we need to consider each *forward pass*, which is a more granular level than each dialogue turn (our temporal resolution of interest in the main text). We denote $\tau$ as the timestep indexing number of token generations. To make the results as general as possible, *i.e.,* encompassing activation steering which acts at each forward pass, we write all results assuming inputs may be introduced during each forward pass $\tau$.

### E.1 Conditions for discrete bottleneck, deterministic system

We start by showing that deterministic LLMs, which produce a distribution over their finite discrete vocabulary space $\Sigma$, will have countable reachable sets.

**Lemma 10** (Deterministic intervened LLMs' reachable sets are countable). *The output attributes of an intervened LLM $\phi$ can be controlled, after $\tau$ token generations, to a set of cardinality at most $\min(|\Sigma|^\tau, |\mathcal{U}|^\tau)$, where $|\Sigma|$ is the vocabulary size.*

*Proof.* We use a cardinality argument. We consider the first token generation. Given any $x_0 \in \mathcal{X}$, fix a $u \in \mathcal{U}$. As $\phi$ is deterministic, we have $|\mathrm{im}(\phi(x_0, u))| = 1$, which implies $|\mathrm{im}(\phi(x_0, \cdot))| \leq |\mathcal{U}|$. Further, $|\mathrm{im}(\phi(\cdot, \cdot))| \leq |\Sigma|$, the vocabulary size. Hence $|\mathrm{im}(\phi(\cdot, \cdot))| \leq \min(|\Sigma|, |\mathcal{U}|)$. Since the readout map $h$ is deterministic, $|\mathrm{im}(h(\cdot, \cdot))| \leq \min(|\Sigma|, |\mathcal{U}|)$. We repeat this reasoning to integrate values of $2 \cdots \tau$, so that $|\mathrm{im}(h(\tau; x_0, u_{1 \cdots \tau}))| \leq \min(|\Sigma|^\tau, |\mathcal{U}|^\tau)$ for all $x_0 \in \mathcal{X}$ and $u_i \in \mathcal{U}$, $i = 1 \cdots \tau$. We have bounded the size of the reachable set for a given $x_0 \in \mathcal{X}$. Then, we apply Def. 15 to see that the dialogue process with deterministic $\phi$ can be controlled to a set of maximum size $\min(|\Sigma|^\tau, |\mathcal{U}|^\tau)$, as the largest intersection between the reachable sets $\mathcal{R}(x_0^{(i)}, \mathcal{U}, \tau)$ for initial states $x_0^{(i)} \in \mathcal{X}$, is less than $\min(|\Sigma|^\tau, |\mathcal{U}|^\tau)$. $\square$

**Lemma 11** (Deterministic, intervened T2IMs' reachable sets have cardinality at most $|\Sigma^*| \times |\mathcal{U}|$). *The output attributes of a deterministic intervened T2IM $\phi$ can be controlled to a set of cardinality at most $|\Sigma^*| \times |\mathcal{U}|$, where $|\Sigma^*|$ is cardinality of all possible input strings.*

*Proof.* (Sketch). We use a similar argument to Lem. 10. If the readout map $h$ (Sec. 3) is deterministic, then the reachable set at time $t = 1$ is only bottlenecked by its initial prompt space $\mathcal{X}_0 = \Sigma^*$ and its input space $\mathcal{U}$. It follows that its reachable set's cardinality is maximum $|\Sigma^*| \times |\mathcal{U}|$, with equality if $h$ is injective. $\square$

This means that for LLMs, there is a guaranteed *discrete bottleneck* from the vocabulary space. For T2IMs, if the input space is continuous, *e.g.,* activation steering, then there is no discrete bottleneck. In our case, as we study discrete-valued inputs of finite length (prompting), there is necessarily a discrete bottleneck for T2IMs.

### E.2 Conditions for discrete bottleneck, stochastic system

Here, we consider under which conditions stochastic systems will have a discrete bottleneck. The discrete bottleneck is guaranteed for LLMs, whose input and output spaces are both strings– however, stochastic T2IMs will have *continuous* output sets.

**Lemma 12** (Stochastic intervened LLMs' reachable sets are countable). *The set of output attributes of an intervened LLM $\phi$ after $\tau$ token generations, have a cardinality at most $|\Sigma|^\tau$, where $|\Sigma|$ is the vocabulary size.*

*Proof.* The number of potential outputs during each forward pass is the number of vocabulary items $|\Sigma|$. After $\tau$ token generations, the maximum number of possible outputs is $|\Sigma|^\tau$. $\square$

**Lemma 13** (Stochastic intervened T2IMs' reachable sets can be uncountable). *The set of output attributes of an intervened T2IM $\phi$ can be uncountable.*

*Proof.* The proof follows from replacing the cardinality of vocabulary space $|\Sigma|$ in Lem. 12 with the cardinality of a T2IM's generation space. As T2IMs generate images, which are continuous, the reachable set of stochastic T2IMs is uncountable. $\square$

### E.3 EXPECTED OUTPUTS CONTROLLABILITY

Stochastic generative models define a distribution that is sampled at inference time. While the output attributes themselves are bottlenecked, their expectations may not be. The expectation of the attribute over the model's distribution is only bottlenecked by the input cardinality $|\mathcal{U}|^\tau$, where $\tau$ is the number of tokens generated in the case of an LLM ($\tau = 1$ for a T2IM).

**Lemma 14** (Stochastic LLMs are expected output controllable on cardinality-$|\mathcal{U}|^\tau$ sets)**.** *The expected measurement value $\mathbb{E}[y]$ of stochastic LMs $\phi$, where the next token $s$ is sampled from $s \sim p_\Sigma$ with $p_\Sigma \in \Delta^{|\Sigma|-1}$,[6] can be controlled to sets of cardinality at most $|\mathcal{U}|^\tau$ by token generation $\tau$.*

*Proof.* We first consider stochastic LLMs. We use the same cardinality argument as in Lem. 10. The proof follows from replacing $\mathrm{im}(\phi) \leq |\Sigma|$ with $\mathrm{im}(\phi) \leq |\Delta^{|\Sigma|-1}| = |\mathbb{R}|$. This means that the expected output attribute $\mathbb{E}_{s \sim p_\Sigma}[y]$ can maximally vary over as many possible values as $p_\Sigma$, which is $|\Delta^{|\Sigma|-1}| = |\mathbb{R}|$. Then, given any $x_0 \in \mathcal{X}$, the cardinality of $\mathrm{im}\left(h(x_0, \cdot)\right)$ is less than $\min(|\mathbb{R}|, |\mathcal{U}|) = |\mathcal{U}|$, where the choice of input $u \in \mathcal{U}$ bottlenecks the expected output attribute. Analogous to the proof of Lem. 10, it follows that, given $x_0$, the reachable set's cardinality $|\mathcal{R}(x_0, \mathcal{U}, \tau)| \leq |\mathcal{U}|^\tau$, by integrating the system. Then, applying Def. 15, the size of the controllable set is maximum $|\mathcal{U}^\tau|$. $\quad\square$

## F REACHABILITY: THEOREM 1

### F.1 THEOREM 1: CONTINUOUS-VALUED VERSION

We state the $\gamma$-quantized version (for continuous-valued measurements) of Thm. 1. The proof for the categorical version follows directly.

---

**Theorem 3: Sample complexity bound, $\gamma$-quantized reachability.**

Let $\mathcal{Y}$ be continuous and bounded. Let $N$ be the covering number of $\mathcal{Y}$ with $\infty$-balls of radius $\gamma/2$. Let $m$ be the number of i.i.d. samples drawn from $Y_t$, *i.e.*, $\{y_i\}_{i=1}^m$ with each $y_i \in \mathcal{Y} \; \forall i$. Fix $\delta \in (0, 1)$. If

$$m \geq \max\left( N, \frac{\log(\delta/N)}{\log(1-p)} \right), \tag{F.7}$$

then $\mathbb{P}(\mathcal{R}_{t,p}^\gamma \subset \cup_{i=1}^m \mathcal{B}_\infty(y_i, \gamma)) \geq 1 - \delta$, where $\mathcal{B}_\infty(y, \gamma)$ is the $l_\infty$ ball centered at $y$ with radius $\gamma$.

---

*Proof.* The problem reduces to asking how many independent samples $m$ are needed to hit all $N$ bins with likelihood $\geq 1 - \delta$, where the probability of sampling each bin is $> p$.

Let $m$ i.i.d. samples be drawn from $p_{y,t}$: $\hat{\mathcal{Y}}^{(m)} = \{y_i\}_{i=1}^m$ (omitting $t$ from the RHS for readability). By definition of $\mathcal{R}_{t,p}^\gamma$, the probability sample $i$ hits the bin $y_{\mathrm{bin}}^j$ in $\mathcal{R}_{t,p}^\gamma$ is $p_{y,t}(y_i \in y_{\mathrm{bin}}^j) \geq p$. The probability sample $i$ lands outside bin $j$ is then $p_{y,t}(y_i \notin y_{\mathrm{bin}}^j) \leq 1 - p$. The probability all samples $i = 1 \cdots m$ land outside nontrivial bin $j$ is $\mathbb{P}(\hat{\mathcal{Y}}^{(m)} \not\subset y_{\mathrm{bin}}^j) \leq (1-p)^m$.

Then, by the union bound over a maximum of $N$ bins, the probability at least one bin is never hit is at most $N(1-p)^m$. Subtracting from 1, the probability that all bins in $\mathcal{R}_{t,p}^\gamma$ are hit by some sample is

$$\mathbb{P}(\text{all } y_{\mathrm{bin}}^j \text{ hit}) \equiv \mathbb{P}(\mathcal{R}_{t,p}^\gamma \text{ is } \gamma\text{-reached by the samples}) \geq 1 - N(1-p)^m. \tag{F.8}$$

We want to guarantee that $\mathbb{P}(\mathcal{R}_{t,p}^\gamma \text{ is } \gamma\text{-reached by the samples}) \geq 1 - \delta$. We do so by guaranteeing $1 - N(1-p)^m \geq 1 - \delta$. Then, solving for $m$, if

$$m \geq \frac{\log(\delta/N)}{\log(1-p)}, \tag{F.9}$$

---

[6]Notation: $\Delta^{|\Sigma|-1}$ is called the "$|\Sigma| - 1$-simplex", or the space of categorical probability distributions over $|\Sigma|$ categories.

then with probability $\geq 1 - \delta$, $\mathcal{R}_{t,p}^\gamma$ is $\gamma$-reached by the samples $\hat{\mathcal{Y}}^{(m)}$. By the pigeonhole principle, $m$ also needs to be greater than the number of bins $N$. It follows that if

$$m \geq \max\left(N, \frac{\log(\delta/N)}{\log(1-p)}\right), \tag{F.10}$$

then all $N$ bins, each with radius $\gamma/2$ (side-length $\gamma$), are hit with probability $\geq 1 - \delta$.

By (F.8), we are now guaranteed w.p. $\geq 1 - \delta$ that each point in $\mathcal{R}_{t,p}^\gamma$ is within $\gamma$ of some sampled point in $\hat{\mathcal{Y}}^{(m)}$. It follows that w.p. $\geq 1 - \delta$, the set $\mathcal{R}_{t,p}^\gamma$ is contained in a $\gamma$-cover of $\hat{\mathcal{Y}}^{(m)}$, or $\mathbb{P}(\mathcal{R}_{t,p}^\gamma \subset \cup_{i=1}^m \mathcal{B}_\infty(y_i, \gamma)) \geq 1 - \delta$.

$\square$

## F.2 COROLLARY OF THEOREM 1: CATEGORICAL VERSION

> **Corollary F.1: Sample complexity bound, categorical reachability**
>
> Let $\mathcal{Y}$ be discrete with finite cardinality $N$. Let $m$ be the number of i.i.d. samples drawn from $Y_t$, *i.e.*, $\{y_i\}_{i=1}^m$ with each $y_i \in \mathcal{Y}$ $\forall i$. Fix $\delta \in (0,1)$. If
>
> $$m \geq \max\left(N, \frac{\log(\delta/N)}{\log(1-p)}\right), \tag{F.11}$$
>
> then $\mathbb{P}(\mathcal{R}_{t,p} \subset \{y_i\}_{i=1}^m) \geq 1 - \delta$.

*Proof.* Follows from proof of Thm. 3. $\square$

## F.3 SAMPLE COMPLEXITY OF THEOREM 1

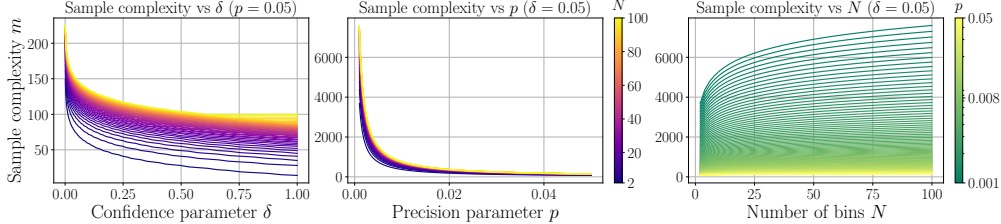

Figure F.14: **Sample complexity growth with hyperparameters.** From left to right, we see sample complexity $m$ growth with respect to $\delta$ (constant $p$), $p$ (constant $\delta$), and $N$ (constant $\delta$). Overall, the most drastic change in sample complexity comes from decreasing $p$. Sample complexity also increases sharply as $\delta$ draws closer to 0. In contrast, sample complexity is robust to increasing $N$ in a fine-grained regime ($N \gtrsim 50$). The sample complexity remains tractable for parameters $(\delta, p, N) = (0.05, 0.05, 100)$, giving $m = 149$.

Figure F.14 visualizes the growth of $m$ with respect to each parameter. The dependence of $m$ on $\delta$ and $p$ is given by the form of Thm. 1, and are seen as a sharp growth as $p$ and $\delta$ grow closer to 0. The dependence on $p$ is more drastic (Figure middle) compared to $\delta$ (Figure left).

The dependence of $m$ on $N$ is less straightforward. We sketch why $m$ grows as $\mathcal{O}(N \log N)$. First, fix the number of bins $N$. Then, $p$ cannot exceed the probability of a bin under uniform density: $p \leq \max_{p_{t,y} \in \mathcal{P}_{t,y}} \min_j p_{t,y}(y_{\text{bin}}^j) = \frac{1}{N}$. We use the observation that $\log(1-p) \approx -p$, valid for small $p \in (0,1)$:

$$\frac{\log(\delta/N)}{\log(1-p)} \approx \frac{\log(N/\delta)}{p} \tag{F.12}$$

$$\geq N(\log N - \log \delta) \tag{F.13}$$

$$= \mathcal{O}(N \log N). \tag{F.14}$$

### F.4 EMPIRICAL VALIDATION OF THEOREM 1

The sample complexity bound in Thm. 1 is distribution-free and holds for any control system. Therefore, the tightness of the bound likely differs across experimental settings, and specifically for different distributions the underlying generative model induces. Therefore, instead of a formal analysis of tightness, which would be highly nontrivial in the absence of the DP's distributional information, in this section we perform a sanity check that probabilistic guarantee in Thm. 1 holds in our setting of interest, *i.e.,* on a generative model.

### F.5 SETUP

We test only the most general case, that is, for a stochastic system $\phi$ and $\gamma$-quantized reachability. In particular, we use Gemma3-4B on the text formality task, for 0-shot prompting and $T = 1$. The initial state is $x_0$=BOS. All other hyperparameters and prompt templates are replicated from Tab. J.4 and J.5 and the input distribution and readout map inherit the formality setting described in the main text.

For the DP defined for the above initial state, input distribution, and readout map, we first proxy the true distribution $p_{y,t}$ by sampling and scoring $N = 10000$ responses from Gemma3-4B. On this proxied true distribution, we compute the "true" $(p, \gamma)$-approximate reachable set $\mathcal{R}_{t,p}^{\gamma}$ as described in Def. 5.

To validate Thm. 1, we run Alg. 1a 200 times. Each time, we check whether the guarantee $\hat{\mathcal{R}}_t^{(m)} \supset \mathcal{R}_{t,p}^{\gamma}$ holds. Then, we average over the 200 runs to obtain the empirical confidence. If the empirical confidence complies with the desired confidence parameter $\delta$, then it serves as a sanity check for Thm. 1.

**Results** Figure F.15 validates Thm. 1 in practice, where we plot the empirical confidence against the sample complexity. Each point on the line represents the mean over 200 runs. Theorem 1 recommends a minimum sample complexity of $m = 104$, which empirically adheres to the guarantee that $\hat{\mathcal{R}}_t^{(m)} \supset \mathcal{R}_{t,p}^{\gamma}$ above the desired confidence $1 - \delta = 0.95$. For this particular setting, Thm. 1 is reasonably tight, given the fewest samples to satisfy the guarantee is around 40. This represents a roughly $2.5\times$ inefficiency in the bound.

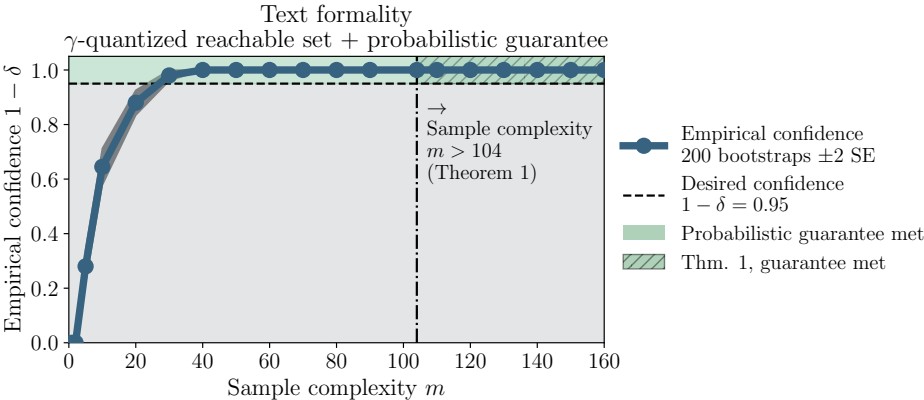

Figure F.15: **Empirical validation of Theorem 1.**

## G EXPECTED OUTPUT REACHABLE SET

Lemma 14 says that considering *expected outputs* under stochastic dynamics no longer suffers from the discrete bottleneck. We present a corollary to DA19's Theorem, (stated in this paper's Thm. 9) that estimates the reachable set of the *expected* output value for stochastic dynamics.

---

**Corollary G.1: Expected output forward reachability for stochastic system**

Let $(\epsilon, \delta) \in (0, 1)$ and let $(\epsilon_\mu, \delta_\mu) \in (0, 1)$. If $m$ satisfies

$$(1 - \delta_\mu)^m (1 - 2n(1 - \frac{\epsilon}{2n})^m) \geq 1 - \delta, \qquad \text{(G.15)}$$

then $\hat{\mathcal{R}}^{(m),y}_{\epsilon_\mu}$, which is the $\hat{\mathcal{R}}^{(m),y}$ of Thm. 9 pushed out by $\epsilon_\mu$, overapproximates an $\epsilon$-accurate reachable set of the *true expected output* with confidence $\delta$, *i.e.*, $\mathbb{P}(\mathcal{R}^{\mathbb{E}Y_1}_{[t_0,t_1],\epsilon} \subset \hat{\mathcal{R}}^{(m),y}_{\epsilon_\mu}) \geq 1 - \delta$.

---

*Proof.* We extend the proof of Thm. 9 to the expectation of $y_t$. The key addition to the proof of Thm. 9 is uncertainty in $\mathbb{E}y_t$ caused by sampling, for which we use a vector variant of Hoeffding's inequality from Pinelis (1994). The inequality states that, for $N$ samples of a random variable $Y \in \mathbb{R}^d$ that has bounded deviations from the mean $\|\overline{Y} - \mathbb{E}Y\|_2 \leq R$, and a choice of $\epsilon_\mu > 0$, then

$$\mathbb{P}(\|\overline{Y} - \mathbb{E}Y\|_2 \geq \epsilon_\mu) \leq \delta_\mu, \qquad \text{(G.16)}$$

where $\delta_\mu = 2\exp\frac{-\epsilon_\mu^2 N}{2R^2} > 0$. That is, for enough samples $N \geq \frac{2R^2}{\epsilon_\mu^2}\log\frac{2}{\delta_\mu}$, $0 < \delta_\mu < 1$, the empirical mean $\overline{Y}$ will approximate the true mean $\mathbb{E}Y$ with high probability.

Hoeffding's inequality enters when choosing the halfspaces $\mathcal{H}_i$ in Thm. 9. Recall that, as $y_t \in \mathbb{R}^n$, there are $2n$ "faces" of the interval covering the reachable set. In Thm. 9, each $\mathcal{H}_i$, $i = 1 \ldots 2n$ is defined as the halfspace facing "away" from the samples in the interval. We modify this definition here. Let $\{\overline{y}_j\}_{j=1}^m$, where $\overline{y}_j = \frac{1}{N}\sum_{k=1}^N y_{j,k}$, be sampled approximations of the true means $\{\mathbb{E}y_j\}_{j=1}^m$. For a choice of $\epsilon_\mu$ and confidence $\delta_\mu$, let $N$ be such that (G.16) holds for all $y_j$; that is, all sampled means $\overline{y}_j$ are within $\epsilon_\mu$ of the true mean $\mu^y_j$, with confidence $\delta_\mu$. Now, define the halfspaces $\mathcal{H}_i$ as the faces of the interval covering all sampled means $\overline{y}_j$, then *shifted each by $\epsilon_\mu$ away from the samples*.

The probability that *all true means* are contained within the bounding box defined by the $\mathcal{H}_i$ is at least the probability that all true means lie within $\epsilon_\mu$ of their corresponding sample. Because the $m$ samples are independent, this probability is given by

$$\mathbb{P}(\cap_{j=1}^m (\|\overline{y}_j - \mu^y_j\|_2 < \epsilon_\mu)) \geq (1 - \delta_\mu)^m. \qquad \text{(G.17)}$$

We now proceed following the proof of Thm. 9, defining the halfspace $\mathcal{P}_i$ with respect to the *true means* $\mu^y_j$, $j = 1 \cdots m$. Let $\hat{\mathcal{R}}^{(m),y}_{\epsilon_\mu}$ be the smallest bounding interval of the sampled means, pushed out by $\epsilon_\mu$. If a $\mu^y_j \in \mathcal{P}_i$, then $\mathcal{H}_i \subset \mathcal{P}_i$, or $p_t(\mathcal{H}_i) \leq p_t(\mathcal{P}_i) = \frac{\epsilon}{2n}$, with probability greater than $(1 - \delta_\mu)^m$. Then, w.p. $(1 - \delta_\mu)^m$, $p_t(\cup_i \mathcal{H}_i) \leq \epsilon \iff p_t(\hat{\mathcal{R}}^{(m),y}_{\epsilon_\mu}) \geq 1 - \epsilon \implies \hat{\mathcal{R}}^{(m),y}_{\epsilon_\mu}$ is an $\epsilon$-accurate reachable set. Overall, let the event $A$ be that $\hat{\mathcal{R}}^{(m),y}_{\epsilon_\mu}$ contains an $\epsilon$-accurate reachable set. Our goal is to bound $\mathbb{P}(A)$. We do so by conditioning $A$ on an event $B$: "$\mathcal{H}_i$ bound the true means". Then, $\mathbb{P}(A) \geq \mathbb{P}(B)\mathbb{P}(A|B)$. The event $A|B$ reduces to the setting in DA19: "the $\mathcal{P}_i$ lie outside the $\mathcal{H}_i$".

$\mathbb{P}(B)$ is at least $(1 - \delta_\mu)^m$. Now, given $B$, that is, given the $\mathcal{H}_i$ bound the true means,

$$\mathbb{P}(A|B) \geq 1 - 2n(1 - \frac{\epsilon}{2n})^m, \qquad \text{(Theorem 1, DA19)} \qquad \text{(G.18)}$$

Then, the overall confidence is given by

$$\mathbb{P}(p_t(\hat{\mathcal{R}}^{(m),y}_{\epsilon_\mu}) \geq 1 - \epsilon) = \mathbb{P}(B)\mathbb{P}(A|B) \geq \underbrace{(1 - \delta_\mu)^m (1 - 2n(1 - \frac{\epsilon}{2n})^m)}_{\text{target probability } 1-\delta}. \qquad \text{(G.19)}$$

If the number of samples $m$ satisfies

$$\underbrace{(1 - \delta_\mu)^m}_{\text{(i) Equation (G.17)}} \underbrace{(1 - 2n(1 - \frac{\epsilon}{2n})^m)}_{\text{(ii) Theorem 1, DA19}} \geq 1 - \delta, \qquad \text{(G.20)}$$

then $\hat{\mathcal{R}}_{\epsilon_\mu}^{(m),y}$ contains the $\epsilon$-accurate reachable set with confidence level $\delta$. $\qquad\square$

Note opposing dependencies of factors **(i)** and **(ii)** on $m$. The probability $\mathcal{P}_i$ do not contain samples *increases* as $m$ increases, while the probability that $\mathcal{H}_i$ do not contain samples *decreases* to 0 as $m$.

# H    CONTROLLABILITY: THEOREM 2

## H.1    OUTPUT CONTROLLABLE SET

**Definition 15** (Output controllable set). Given a control system $(\phi, \mathcal{X}, \mathcal{U}, \mathcal{T}, y)$, the output controllable set $\mathcal{C}_t \subseteq \mathcal{Y}$ at time $t$ is given by

$$\mathcal{C}_t = \bigcap_{x_0 \in \mathcal{X}_0} \mathcal{R}_t(x_0, \mathcal{U}), \tag{H.21}$$

or the intersection of the reachable sets of all initial states $x_0 \in \mathcal{X}_0$.

## H.2    PROOF OF THEOREM 2

We state the proof of Thm. 2 for the quantized case. The proof of the categorical case immediately follows by removing $\gamma$ everywhere it is written.

*Proof.* Let $E$ be the event that $\mu(\hat{\mathcal{C}}_t \setminus \mathcal{C}_t^\alpha) < \epsilon$. We want to $\mathbb{P}(E)$ to be greater than some $1 - \delta$. Now, let $F$ be the event that $\mathcal{R}_{t,p}^\gamma(x_0^{(i)}) \subset \hat{\mathcal{R}}_t(x_0^{(i)})$ for all $i = 1 \ldots k$. That is, $F$ is the event that all sampled reachable sets contain the true target. We have $\mathbb{P}(E) \geq \mathbb{P}(E|F)\mathbb{P}(F)$. Our strategy will be to lower bound the RHS by lower-bounding each term on the RHS individually.

We start with $\mathbb{P}(F)$. By independence of the $k$ sampled reachable sets, we have $\mathbb{P}(F) \geq (1 - \delta_R)^k$, by Thm. 3.

We now consider the term $\mathbb{P}(E|F)$. Abbreviate $\hat{\mathcal{C}}_k = \cap_{i=1}^k \hat{\mathcal{R}}_t(x_0^{(i)})$ to be the empirical intersection of $k$ sampled sets, where each $x_0^{(i)} \sim p_0$, $i = 1 \ldots k$. We aim to bound the error $\mu(\hat{\mathcal{C}}_k \setminus \mathcal{C}_t^\alpha)$. For brevity, we rewrite $B = \hat{\mathcal{C}}_k \setminus \mathcal{C}_t^\alpha$ to mean all points outside the true $\mathcal{C}_t^\alpha$, but inside the empirical $\mathcal{C}_k$. All points $y_{\text{bin}}$ in the true $(\alpha)$-controllable set $\mathcal{C}_t^\alpha$ have $\mu(y_{\text{bin}}) > 1 - \alpha$. All points in $B$, are *bins* that have *incorrectly survived $k$ samples*. In other words, each element $y_{\text{bin}}$ of $B$ is such that $\mu(y_{\text{bin}}) < 1 - \alpha$. Our aim is to find a number of sampled sets $k$ after which

$$\mathbb{P}(\mu(B) < \epsilon \mid F) \geq 1 - \delta_C. \tag{H.22}$$

This is achieved with Markov's inequality:

$$\mathbb{P}(\mu(B) > \epsilon \mid F) \leq \frac{\mathbb{E}[\mu(B) \mid F]}{\epsilon}. \tag{H.23}$$

Then, if $\mathbb{E}[\mu(B) \mid F]/\epsilon \leq \delta_C$, then (H.22) holds. Note that $\mathbb{E}[\mu(B) \mid F]$ is equal to the probability a $y_{\text{bin}}$ with $\mu(y_{\text{bin}}) < 1 - \alpha$ is hit by $k$ samples, given that each sampled reachable set is accurate. This likelihood is at most

$$\mathbb{E}[\mu(B) \mid F] = \mathbb{P}(y_{\text{bin}} : \mu(y_{\text{bin}}) < 1 - \alpha \text{ hit by } k \text{ samples} \mid \text{all reachable set samples accurate}) \tag{H.24}$$

$$\leq (1 - \alpha)^k. \tag{H.25}$$

Then, if

$$\frac{(1 - \alpha)^k}{\epsilon} \leq \delta_C \tag{H.26}$$

$$\implies k \geq \frac{\log \epsilon \delta_C}{\log(1 - \alpha)}, \tag{H.27}$$

then $\mathbb{P}(\mu(B) < \epsilon \mid F) \geq 1 - \delta_C$, as desired.

Putting the two terms together, we have that if $k \geq \frac{\log \epsilon \delta_C}{\log(1-\alpha)}$, then with probability at least $(1 - \delta_C)(1 - \delta_R)^k$, $\mu(\hat{\mathcal{C}}_t \setminus \mathcal{C}_t^\alpha) \leq \epsilon$. $\qquad\square$

## H.3 AUTO-PARAMETER SELECTION FOR THEOREM 2

There are several free parameters in Thm. 1 and 2. If the end goal is a *controllability* analysis, then given a target confidence $1 - \delta \leq (1 - \delta_R)^k (1 - \delta_C)$, we can compute the optimal values for $\delta_R$ and $\delta_C$ such that the total sample complexity $n = m \cdot k$ is minimized. The intuition is that $\delta_R$ and $\delta_C$ are a tradeoff between more reachable set samples ($m$) and more initial state samples ($k$).

Formally, this can be written as the following constrained optimization problem:

$$\text{minimize} \quad n = m \cdot k \quad \text{(Total number of samples)} \tag{H.28}$$

$$\text{subject to} \quad m \geq \frac{\log(\delta_R/N)}{\log(1-p)} \quad \text{(Reachability, Thm. 1)} \tag{H.29}$$

$$k \geq \frac{\log(\epsilon \cdot \delta_C)}{\log(1-\alpha)} \quad \text{(Controllability, Thm. 2)} \tag{H.30}$$

$$(1-\delta_R)^k(1-\delta_C) \geq 1 - \delta \quad \text{(Overall confidence constraint)} \tag{H.31}$$

$$\text{with fixed parameters} \quad \delta, \ p, \ \alpha, \ \epsilon \in (0,1) \tag{H.32}$$

$$N \in \mathbb{N}_{\geq 2} \tag{H.33}$$

$$\text{and optimization variables} \quad m, \ k \in \mathbb{N}_{\geq 2}, \quad \delta_C, \ \delta_R \in (0,1). \tag{H.34}$$

If we only require an overall confidence $\delta$ on the controllable set (RHS of Theorem 2), we can abstract away the decision of $\delta_R$ and $\delta_C$. By doing a dense grid search over $(\delta_C, \delta_R)$ on `np.geomspace(1e-6, 0.999, 250)`$^2$, we can find the minimum total number of samples needed to satisfy all the constraints in Thm. 1 and 2 and (H.31). The extra time taken by this step is negligible compared to the time it takes to sample the dialogue process.

## H.4 SAMPLE COMPLEXITY OF THEOREM 2

We split the analysis into two parts: **(i)** dependence of $k$ on hyperparameters, see Fig. H.16, then **(ii)** dependence of the total sample complexity $n = m \cdot k$ on hyperparameters, using the subroutine described in App. H.3.

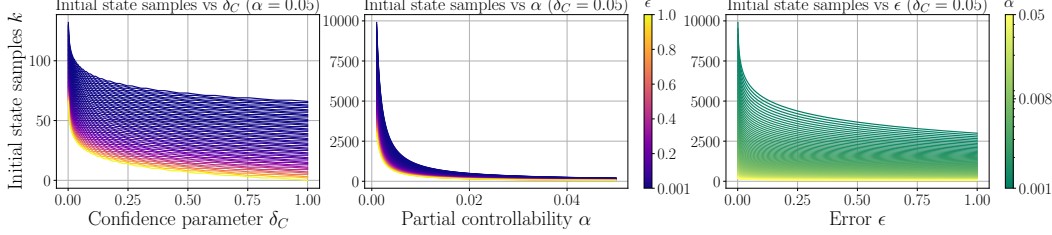

Figure H.16: **Number of initial state samples dependence on hyperparameters.** From left to right, we see the number of initial state samples $k$ grow with respect to $\delta_C$ (constant $\alpha$), $\alpha$ (constant $\delta_C$), and $\epsilon$ (constant $\delta_C$). Overall, the most drastic change in sample complexity comes from decreasing $\alpha$, the partial controllability parameter. Sample complexity also increases sharply as $\delta_C$ approaches 0. In contrast, sample complexity is robust to increasing $\epsilon$, for $\epsilon$ closer to 1, especially for more lenient partial controllability (higher $\alpha$).

Figure H.16 shows that the number of sampled initial states highly depends on $\alpha$ (middle panel), while comparatively robust to the confidence parameter $\delta_C$ and error $\epsilon$. What is more important is the *total number of samples* $n$. We performed the procedure in App. H.3 to abstract away the choice of $\delta_C$ and $\delta_R$, and analyzed the dependence of total sample complexity $n$ on $p$, $N$, $\epsilon$, and $\alpha$.

Figure H.17 shows how $n$ varies with each hyperparameter. Sample complexities hover near $\lesssim 10^7$ for very small ($\approx 0.001$) values of $\alpha$ and $p$.

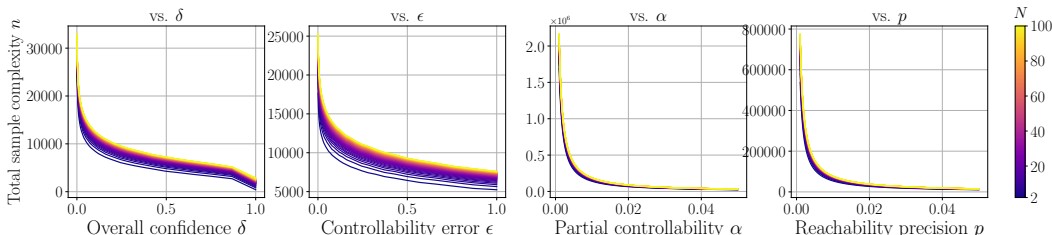

Figure H.17: **Total sample complexity as a function of hyperparameters.** We vary the total number of samples $n$ in controllable set estimation as a function of each fixed hyperparameter $(\delta, \epsilon, \alpha, p, N)$, one at a time. When varying each parameter (each panel), we fix all others except for $N$, which varies, at a constant value of 0.05. Given overall confidence $\delta$, we auto-select $\delta_C$ and $\delta_R$ using the algorithm in App. H.3.

## I    HYPOTHESIS TESTING

Because Thm. 1 and 2 estimate confidence sets, they lend themselves to hypothesis testing. We give an intuition in the below example for the reachable set:

---

**Example I.1: Hypothesis testing, reachable set**

1. **Q:** I have a toxicity verifier. Is it possible to control my LLM's toxicity to $[0, 0.4]$ with prompting? I am fine with an error of $\gamma = 0.05$, and I consider a region to have nontrivial probability if its probability is greater than $p = 0.001$.

2. **Quantize.** Estimate $N = 0.4/0.05 = 8$ bins.

3. **Null hypothesis**: My target range $[0, 0.4]$ is 0.05-quantized reachable.

4. Set the $p$-value cutoff to, *e.g.,* $\delta = 0.05$.

5. Sample $m \geq 2204$ toxicity scores (applying Thm. 1).

6. Suppose we obtain $\hat{\mathcal{R}}_t^{(m)} = [0, 0.07] \cup [0.5, 0.8] \not\supset \mathcal{Y}^*$.

7. Reject the null hypothesis, concluding that $\mathcal{Y}^*$ is *not* reachable within $\gamma$ with a $p$-value cutoff $\delta$.

---

## J    EXTENDED EXPERIMENTAL SETUP

Table J.4: **Hyperparameters for all controllability experiments.**

| Variable | Description | Value |
|---|---|---|
| $\alpha$ | partial controllability, Thm. 2 | 0.1 |
| $\epsilon$ | controllable set error, Thm. 2 | 0.05 |
| $\delta$ | controllable set confidence, Thm. 2 | 0.05 |
| $p$ | reachability precision, Thm. 1 | 0.05 |
| $\gamma$ | quantization, Thm. 1 | 0.1 |

Table J.5: **Model sampling hyperparameters.** For the stochastic setting, *i.e.,* where we sample the underlying LLM, as well as for initial state generation $p_0$ with Mistral, we use the following sampling parameters.

| | |
|---|---|
| `max_new_tokens` | 100 |
| temperature (stochastic $\phi$) | 0.7 |
| temperature ($p_0$) | 0.9 |
| `top_p` (stochastic $\phi$) | 0.9 |
| `top_p` ($p_0$) | 0.95 |
| `top_k` | 50 |

### J.1    INPUT DISTRIBUTIONS

We here describe the input distributions related to each task in the experimental results.

**LLM task templates**

```
──────────── Generate an even/odd number | Template 0-shot ────────────
"Generate a strictly positive {} integer."
```

```
──────────── Generate an even/odd number | Template 5-shot ────────────
"""
You are a chatbot that follows instructions. I will give you some examples:
    Instruction: Generate a strictly positive odd integer. Only generate the integer and
    ↪    nothing else.
```

```
    Result:
    31

    Instruction: Generate a strictly positive even integer. Only generate the integer
↪   and nothing else.
    Result:
    94726

    Instruction: Generate a strictly positive odd integer. Only generate the integer and
↪   nothing else.
    Result:
    7

    Instruction: Generate a strictly positive even integer. Only generate the integer
↪   and nothing else.
    Result:
    200

    Instruction: Generate a strictly positive odd integer. Only generate the integer and
↪   nothing else.
    Result:
    2399

    Knowing this, please follow the next instruction.

    Instruction: Generate a strictly positive {} integer. Only generate the integer and
↪   nothing else.
    """
```

─────────────── Generate an even/odd number | Feedback template ───────────────
```
"Your answer was incorrect. I asked for a {desired_value} number, and you gave an
↪   {last_str}. Please try again. Strictly write the integer without quotes and do not write
↪   anything else."
```

─────────────── Generate number of characters | Template 0-shot ───────────────
```
"Write a string of {} characters. Strictly write the string without quotes and do not write
↪   anything else."
```

─────────────── Generate number of characters | Template 0-shot ───────────────
```
"""
You are a chatbot that follows instructions. I will give you some examples:

        Instruction: Write a string of 2 characters.
        Result:
        Hi

        Instruction: Write a string of 5 characters.
        Result:
        Hello

        Instruction: Write a string of 7 characters.
        Result:
        abcdefg

        Instruction: Write a string of 4 characters.
        Result:
        Cars

        Instruction: Write a string of 10 characters.
        classifier

    Knowing this, please follow the next instruction.

    Instruction: Write a string of {} characters. Strictly write the string without quotes
↪   and do not write anything else.
    """
```

─────────────── Generate number of characters | Feedback template ───────────────
```
"Your answer was too {custom_str}. I asked for a string consisting of {desired_value}
↪  characters, and you produced one with {last_outputs} characters. Please try again.
↪  Strictly write the string without quotes and do not write anything else."
```

─────────────────────── Generate formality | Template 0-shot ───────────────────────
```
"Generate a story that is {:.2f}\% formal. Write the story and only the story."
```

─────────────────────── Generate formality | Template 5-shot ───────────────────────
```
"""
You are a chatbot that follows instructions. I will give you an example:
    Instruction: Generate a story that is 100% formal.
    Result:
    Kind sir, please make your way to the terrace.

    Instruction: Generate a story that is 75% formal.
    Result:
    She bought the book, then she went to the beach.

    Instruction: Generate a story that is 50% formal.
    Result:
    Hey! Just went to the store.

    Instruction: Generate a story that is 25% formal.
    Result:
    Wow I'm so tired.

    Instruction: Generate a story that is 0% formal.
    Result:
    hey man how's it goin

    Knowing this, please follow the next instruction.

    Generate a story that is {}\% formal. Write the story and only the story.
    """
```

─────────────────── Generate formality | Feedback template ───────────────────
```
"Your answer was too {formal_str}. I asked for a formality level of {desired_value:.2f}, and
↪  you gave a sentence with formality {100 * last_outputs:.2f}. Please try again. Write the
↪  story and only the story."
```

─────────────────────── Generate word length | Template 0-shot ───────────────────────
```
"Generate a sentence where the average word length is exactly {} letters. Strictly write the
↪  sentence without quotes and do not write anything else."
```

─────────────────────── Generate word length | Template 5-shot ───────────────────────
```
"""
 You are a chatbot that follows instructions. I will give you an example:

    Instruction: Generate a sentence where the average word length is exactly 4 letters.
    Result:
    The pig jumped above her.

    Instruction: Generate a sentence where the average word length is exactly 3.7 letters.
    Result:
    She bought the book, then she went to the beach.

    Instruction: Generate a sentence where the average word length is exactly 3.5 letters.
    Result:
    Hey! Just went to the store.

    Instruction: Generate a sentence where the average word length is exactly 3 letters.
    Result:
    Wow I'm so tired.

    Instruction: Generate a sentence where the average word length is exactly 10 letters.
```

```
    Result:
    Stop redistribution!

    Knowing this, please follow the next instruction.

    Generate a sentence where the average word length is exactly {} letters. Strictly write
↪   the sentence without quotes and do not write anything else.
    """
```

```
─────────────── Generate word length | Feedback template ───────────────
"Your answer was too {custom_str}. I asked for a string whose average word length is
↪   {desired_value:.2f} letters, and you produced one with average word length
↪   {last_outputs:.2f}. Please try again. Strictly write the sentence without quotes and do
↪   not write anything else."
```

**Text-to-Image task templates**

```
─────────────── Generate N objects | Template 0-shot ───────────────
"White background. {} {object}."
# object choices: COCO classes
```

```
─────────────── Generate object at position | Template 0-shot ───────────────
"White background. A {object} at the {pos} of the image."
# object choices: COCO classes
# pos choices: ["top left", "top right",  "bottom left", "bottom right", "center"]
```

```
─────────────── Generate saturation | Template 0-shot ───────────────
"{:.2f}\% saturation."
```

## K  CONTROLLABILITY PACKAGE

We open-source a Python package to test controllability and estimate reachable sets. The package is completely built on PyTorch and supports loading of local or HuggingFace models to realize initial state or input distributions, as well as the readout map. An example usage is given below:

```
─────────────── Basic usage of our toolkit ───────────────
import torch
from controllability.systems.control_system import ControlSystem
from controllability.verifiers.reachability import Reachability
from controllability.verifiers.controllability import Controllability

# Define initial states, input distributions
initial_states_distribution = ...
input_distribution = ...

# Define control system
dialogue_process = ControlSystem.from_model_name(
    "google/gemma-3-4b-it",
    output_map="huggingface/text-classifier",
    output_space=[[0, 1]],
    input_distribution=input_distribution,
    # model_config= {do_sample=True, ...}
)

# Reachability experiment
time_horizon = 5
reachability_verifier = Reachability.from_problem_type(
    "quantized",
    dialogue_process,
    input_distribution,
    p=0.05,
    delta=0.05
```

```
)

reachable_set = reachability_verifier.get_reachable_set(
    initial_state="Hello! ",
    time_horizon
)

# Controllability experiment
controllability_verifier = Controllability.from_reachability_problem(
    reachability_verifier,
    epsilon=0.05,
    alpha=0.1,
    delta=0.05
)
controllable_set = controllability_verifier.get_controllable_set(
    initial_states_distribution,
    time_horizon
)
```

## L    USE OF AI WRITING ASSISTANCE

We acknowledge the use of a large language model (LLM) to assist in refining the language, grammar, and clarity of this manuscript. All content was originally drafted by the human authors, and all AI-generated suggestions were critically reviewed, edited, and approved by the authors, who retain full responsibility for the final text.

