# OpenReview forum: "GenCtrl -- A Formal Controllability Toolkit for Generative Models"
_ICLR.cc/2026/Conference — ICLR 2026 Poster_

### Official Review · Reviewer_vGbb · 2025-10-31

**Soundness:** 3
**Presentation:** 3
**Contribution:** 3
**Rating:** 8
**Confidence:** 3

**Summary:**

This paper establishes a theoretical foundation for controllable generative modeling from a control-theoretic perspective, framing generation controllability in terms of the classic concepts of reachability and controllability. It provides a rigorous theoretical proof to derive the conditions for controllability. Experimental results ultimately demonstrate that current generative models are fragile in terms of their controllability.

**Strengths:**

1. The paper introduces a control-theoretic framework to formalize the controllability of generative models. The theoretical analysis yields specific conditions for achieving controllability, offering valuable foundational support for advancing controllability algorithms and control input design.
2. Through evaluations across different task configurations and various LLMs, the paper demonstrates that controllability performance varies significantly with both the task and the specific LLM employed.

**Weaknesses:**

1. According to the Kalman controllability rank condition, the controllability of a dynamical system is determined by its intrinsic properties. For an LLM, if its inherent capability in a certain attribute is weak, we can postulate that its controllability regarding that attribute will consequently be poor. For instance, a 3B parameter model inherently possesses limited mathematical and reasoning capabilities. Therefore, if evaluated on larger models with significantly enhanced overall abilities, does the authors' conclusion that "LLM controllability is fragile" still hold?
2. As noted in reference [1], if the attributes manifested in the LLM's input are opposite to the target attributes, a substantial and longer control input is required for correction, which also reflects weaker controllability. Thus, in the experimental setup, should the authors clarify the input conditions and endeavor to avoid inputs with such opposing attributes?

[1] Fundamental Limitations of Alignment in Large Language Models

**Questions:**

See the weakness.

---

> ### Author Response · Authors · 2025-11-21
> **Response**
>
> Thanks for your suggestions and encouraging review!
>
> >  For an LLM, if its inherent capability in a certain attribute is weak, we can postulate that its controllability regarding that attribute will consequently be poor. For instance, a 3B parameter model inherently possesses limited mathematical and reasoning capabilities. Therefore, if evaluated on larger models with significantly enhanced overall abilities, does the authors' conclusion that "LLM controllability is fragile" still hold?
>
> This is a great point. We added the experiment to the manuscript. It is true that the expressivity of the dynamics impacts the controllability of the system. We ran new experiments testing the controllability on the Qwen3 family, with sizes ranging from 0.6B to 14B, in a 0-shot 5-turn setting, for the tasks of formality, string length and average word length. We found that larger models were generally more controllable, but that calibration quickly plateaus at ~4B-8B. The exact degree of controllability and calibration are still highly task-dependent, however (Ex., controllability is great for formality and worse for string length and average word length; calibration was bad for formality, and better for string length and average word length). Please see updated Section 6.1 at line 426 “Larger models are more controllable”, updated Figure 3 in the main body, and A.8, A.9.
>
> > As noted in reference [1], if the attributes manifested in the LLM's input are opposite to the target attributes, a substantial and longer control input is required for correction, which also reflects weaker controllability. Thus, in the experimental setup, should the authors clarify the input conditions and endeavor to avoid inputs with such opposing attributes?
>
> We agree, and it’s a very good point. We added the citation and short discussion on page 7 where we discuss the inputs to the model. In our experimental setup, we made sure that our initial states are completely unrelated to the tasks (i.e., our initial states are conversation openers to the LLM). However, our framework can be used to perform statistical hypothesis testing (see Appendix I for an example) for questions like yours, where the authors in [1] could validate their findings with guarantees, and observe that maybe longer dialogue horizons lead to better controllability in the case of opposing inputs.

---

> > ### Author Response · Authors · 2025-11-28
> > **Rebuttal Follow-Up**
> >
> > Dear Reviewer vGbb,
> >
> > Thank you for your encouraging review and high rating. We have carefully addressed your points in our rebuttal.
> >
> > To briefly summarize:
> >
> > * We conducted new experiments on the Qwen3 family (0.6B to 14B), showing larger models are generally more controllable but calibration plateaus, and task dependence remains. These results are now in Section 6.1, Figure 3, and Appendix A.8, A.9.
> > * We clarified our experimental input conditions, ensuring initial states were task-unrelated, and highlighted how our framework can rigorously test hypotheses regarding opposing attributes.
> >
> > We hope these additions and clarifications have further strengthened the paper. We would be grateful for any updated thoughts on our submission, particularly considering the extensive revisions and new evidence provided.
> >
> > Thank you for your time and expertise.
> >
> > Sincerely,
> >
> > Authors

---

### Official Review · Reviewer_N39G · 2025-11-01

**Soundness:** 4
**Presentation:** 4
**Contribution:** 3
**Rating:** 6
**Confidence:** 2

**Summary:**

This paper provides an interesting stand - it formalizes user-model interaction as a control process, then gives PAC-style sample-complexity bounds for estimating reachable and (partially) controllable output sets of black-box generative models. A pair of MC algorithms (reachability and controllability) come with finite-sample guarantees. Experiments across a large variety of models suggest that, the controllability that we usually depend on is fragile and highly task/model dependent.

**Strengths:**

1) The formalization of dialogue-as-control is clear and rigorous.

2) The paper provides PAC-style, distribution-free bounds, avoiding intractable verification of global smoothness, thus suits black-box examinations well.

3) The proposed algorithms are simple and easy to follow.

4) The eventually alerting conclusion reveals the fragility of controllability in LLMs to the community, which could be of higher impacts.

**Weaknesses:**

1) Please correct me if I misunderstood this: how about when N -> \infty? In practice this is easy due to the dimensionality curse, and thus the total demanded sampling number by THM1 can explode to some extent.

2) The role of the readout h is a little bit less clearly demonstrated. In my understanding, isn't it the case that its calibration can be very hard in practice when considering the general case? I would appreciate if the authors can provide guidance for me to better understand how this part can work robustly.

**Questions:**

Please refer to Weakness).

---

> ### Author Response · Authors · 2025-11-21
> **Response**
>
> Thanks for your helpful comments! We hope to clarify your questions below.
>
> > how about when N -> \infty? In practice this is easy due to the dimensionality curse, and thus the total demanded sampling number by THM1 can explode to some extent.
>
> Thanks for allowing us to clarify this point. In Theorem 1 we require $N$ to be finite, as we mention in the Theorem box, L250 (“Let $\mathcal{Y}_q$ have finite cardinality $N$”). Additionally, we also emphasize the boundedness requirement in the abstract (“employ no assumptions except for output boundedness”) and in L231 (“i.e., that $\mathcal{Y}$ is bounded”. We will include a comment in the contributions section of the introduction, mentioning in L57: “[...] as they are distribution-free, only assume boundedness of the target attribute, and explicitly account for the [...]”..
>
> Please see Appendix F.3 for a discussion on how sample complexity grows with $N$. We have added to that Discussion and Limitations section the below discussion of the high-dimensional case, where d attributes are being jointly controlled at once. Let us know if anything remains unclear!
>
> “In Theorem 1, the sample complexity to estimate the reachable set scales with its covering number $N$. When estimating joint reachability over $d$ attributes, the covering number $N$ of the reachable set grows not with the attribute space's extrinsic dimension $d$, but rather its intrinsic dimension [1], which is typically lower in practice. This alleviates sample complexity explosion due to the curse of dimensionality to some extent. Still, Theorem 1 does not scale well when estimating intrinsically complex reachable sets with high precision. This remains an important open problem not only for our setting, but for high-dimensional reachable set estimation in general (Bansal and Tomlin, 2020; Lin and Bansal, 2023; Devonport and Arcak, 2020). As a workaround, we recommend constructing a quantization $\mathcal Y_q$ of the attribute space with tractable cardinality $N$; for example, constructing $\mathcal Y_q$ to correspond to binary safe vs.~unsafe regions of attribute space (this sets $N=2$).”
>
> > The role of the readout h is a little bit less clearly demonstrated. In my understanding, isn't it the case that its calibration can be very hard in practice when considering the general case?
>
> Thanks for bringing this point up, and allowing us to clarify. The readout $h$ is a function that maps the output of the generative model to some attribute, like string length or text formality (see line 145 and Figure 1). For example, in the case of string length, $h$ can be given by Python’s `len()` function. We are estimating the controllability of the attribute as measured by $h$. We have added this string length example around line 145 in the updated manuscript.
>
> In our framework, the practitioner is in charge of designing $h$ and making sure it captures the intended task semantics. For instance, if a practitioner wants to control the sentiment of a model, they should make sure $h$ is a good sentiment classifier. However, how $h$ might be calibrated to task semantics, e.g., making sure $h$ is a good sentiment classifier, is not part of our analysis. We believe there might have been a confusion with the calibration metrics and analysis we provide in Section 6, which measure whether the *LLM/T2IM* (not $h$ per-se) is calibrated with respect to the user requests.

---

> ### Comment · Reviewer_N39G · 2025-11-28
> **Thanks for the rebuttal**
>
> I have clearly read the response and found them satisfactory. I still incline to moderately support the acceptance, and have increased my confidence in appreciation of the authors' response. The changes will take effect whenever the system allows it.

---

> ### Author Response · Authors · 2025-11-28
> **Grateful for score increase and valuable suggestions**
>
> Dear Reviewer N39G,
>
> Thank you for engaging with our rebuttal and for explicitly stating that you found our responses satisfactory, leading to an increase in your confidence. We truly appreciate your thorough review and constructive feedback.
>
> Given that your concerns regarding the sample complexity and the role of h have been addressed, we were wondering if there are any remaining aspects or points that, if further clarified or elaborated upon, would help you reconsider your overall assessment of our submission?
>
> Thank you for your continued time and consideration.
>
> Sincerely,
>
> Authors

---

### Official Review · Reviewer_ANgd · 2025-11-02

**Soundness:** 3
**Presentation:** 4
**Contribution:** 3
**Rating:** 6
**Confidence:** 3

**Summary:**

This paper proposes a controllability framework to assess whether black-box generative models are controllable prior to designing control mechanisms. The authors model user-model interaction as a control process and introduce Monte Carlo algorithms to estimate the set of achievable outputs. These algorithms are supported by probably-approximately correct (PAC) guarantees. Applying this framework to several LLMs and T2IMs, the paper concludes that controllability is a fragile, task-dependent property, challenging the implicit assumption that these models are generally controllable.

**Strengths:**

1. The central idea of shifting focus from how to control models to if they can be controlled is a valuable and novel contribution. It brings a much-needed layer of formal scrutiny to a field dominated by empirical trial-and-error.

2. The paper is well-written, and the authors do an admirable job of introducing concepts from control theory and adapting them to the context of generative models, particularly with their discussion of the discrete bottleneck.

**Weaknesses:**

1. The paper claims its framework can assess any control mechanism, including finetuning and representation engineering. Yet, the experiments are confined exclusively to prompting, arguably the weakest and least reliable form of control.

2. The framework is not scalable to complex, high-dimensional control problems. The sample complexity scales with the number of discretized output bins, N. For any realistic task involving the control of multiple attributes simultaneously (e.g., style, tone, and factual accuracy), the resulting number of bins would become combinatorially explosive, making the method computationally infeasible.

3. Furthermore, the analysis is purely descriptive; it tells us that a model is uncontrollable for a given task, but provides no diagnostic insight into why it fails or how one might improve the system. This severely limits its practical value beyond a simple and potentially misleading assessment.

**Questions:**

Please see the weakness section.

---

> ### Author Response · Authors · 2025-11-21
> **Response**
>
> Thanks for your feedback! Let us know if anything remains unclear after the response.
>
> > The paper claims its framework can assess any control mechanism, including finetuning and representation engineering. Yet, the experiments are confined exclusively to prompting, arguably the weakest and least reliable form of control.
>
> Thanks for allowing us to clarify this point. Our paper is primarily a theoretical contribution that we then showcase on prompting (albeit on many diverse tasks, prompting methods, models, and both vision and language). We demonstrated our theory on prompting due to widespread user adoption and ease of application to vision and language models. However, our theory admits any and all control mechanisms $u$ whose distribution $p_U$ can be sampled. This follows immediately from the proof of Theorem 1, which does not use any information whatsoever about $u$. Indeed, our intention with this paper is not to compare possible control mechanisms, but rather to provide a rigorous tool that practitioners can universally apply to their use-case.
>
> > The framework is not scalable to complex, high-dimensional control problems. The sample complexity scales with the number of discretized output bins, N. For any realistic task involving the control of multiple attributes simultaneously (e.g., style, tone, and factual accuracy), the resulting number of bins would become combinatorially explosive, making the method computationally infeasible.
>
> Thanks for raising this point about the curse of dimensionality, where the covering number $N$ appears to grow exponentially in the number of output attributes. This applies when trying to jointly control $d$ attributes, and is an issue that any classic sampling method might encounter (and happens to be an open problem in the high-dimensional reachability estimation literature, which considers d=10 to be high dimensional! [1]). We have added the following to the discussion (this was missing, thanks for pointing it out).
>
> In Theorem 1, the sample complexity to estimate the reachable set scales with its covering number $N$. When estimating joint reachability over $d$ attributes, the covering number $N$ of the reachable set grows not with the attribute space's extrinsic dimension $d$, but rather its intrinsic dimension [1], which is typically lower in practice. This alleviates sample complexity explosion due to the curse of dimensionality to some extent. Still, Theorem 1 does not scale well when estimating intrinsically complex reachable sets with high precision. This remains an important open problem not only for our setting, but for high-dimensional reachable set estimation in general [2-4]. As a workaround, we recommend constructing a quantization $\mathcal Y_q$ of the attribute space with tractable cardinality $N$; for example, constructing $\mathcal Y_q$ to correspond to binary safe vs.~unsafe regions of attribute space (this sets $N=2$).
>
> Finally, at this moment in the literature on LLM/T2IM attribute control, the vast majority of works focus on controlling single attributes, achieving compositionality with limited success [5, 6, 7]. We maintain that our framework, tractable in these cases, is a necessary and useful first step for the community before scaling to high dimensions.
>
>
> [1] Intrinsic Dimension Estimation Using Packing Numbers (Kegl, NeurIPS 2002)
>
> [2] DeepReach (Bansal and Tomlin, ICRA 2020)
>
> [3] Estimating reachable sets with scenario optimization. (Devonport and Arcak, Conf. on Learning for Dynamics and Control 2020)
>
> [4] Verification of neural reachable tubes via scenario optimization and conformal prediction. (Lin and Bansal, Conference on Learning for Dynamics & Control, 2023)
>
> [5] Progressive Compositionality in Text-to-Image Generative Models. (Xu Han et al. 2025)
>
> [6] How Many Instructions Can LLMs Follow at Once? (Jaroslawicz et al. 2025)
>
> [7] T2I-CompBench++: An Enhanced and Comprehensive Benchmark for Compositional Text-to-image Generation. (Huang et al. 2025)

---

> ### Author Response · Authors · 2025-11-21
> **Response cont'd**
>
> > no diagnostic insight into why it fails or how one might improve the system
>
> As an analogy, consider training a text classifier and measuring precision-recall, which is a generic measure that tells the quality of the classifier. In the case where precision-recall is low, one would go back and apply diagnostic tools specific to their use-case to interpret and fix the issue. Our method is akin to the “precision-recall” in this example, where we provide the controllable set of the LLM with formal guarantees.
>
> Causal or mechanistic analysis of controllability failures, as well as controller design (see line 164), is not in the scope of our work. Our framework does however allow a transparent error analysis, since Algs 1 and 2 return all sampled trajectories including all input-output mappings (as in Section 6). This can be of great use to practitioners to analyze what causes controllability failures for their specific use-cases. We added this to the Limitations and Conclusion.
>
> > This severely limits its practical value beyond a simple and potentially misleading assessment.
>
> Thanks for this point. We kindly disagree that our method produces potentially misleading assessments. We provide a framework that rigorously defines and quantifies controllability, providing controllability error estimates with formal guarantees (Theorems 1 and 2).
>
> To the best of our knowledge, our work is the first controllability accounting method to be introduced for generative models.
> We foresee all kinds of potential uses in generative model safety and compliance, including rigorously comparing different control mechanisms, estimating reachable sets under adversarial inputs, enforcing controllability during training, and accounting in policy and deployment. This is now added to the Conclusion.

---

> ### Author Response · Authors · 2025-11-28
> **Rebuttal Follow-Up**
>
> Dear Reviewer ANgd,
>
> Thank you for your constructive feedback and for recognizing the novelty of our approach. We have carefully addressed your concerns in our rebuttal.
>
> To briefly summarize:
>
> * We clarified that our framework is a theoretical contribution applicable to any control mechanism, with prompting used as a practical demonstration.
> * We addressed scalability for high-dimensional problems by discussing intrinsic dimension and practical workarounds like attribute quantization, noting its current utility for single-attribute control.
> * We explained our method as a rigorous measurement tool for controllability, providing data for practitioners to diagnose failures, rather than designing controllers or providing causal explanations.
>
> We hope these additions and clarifications have fully addressed your points. We would be grateful for any updated thoughts on our submission, particularly considering the extensive revisions and new evidence provided.
>
> Thank you for your time and expertise.
>
> Sincerely,
>
> Authors

---

### Official Review · Reviewer_A4fa · 2025-11-04

**Soundness:** 3
**Presentation:** 3
**Contribution:** 3
**Rating:** 6
**Confidence:** 3

**Summary:**

This paper provides a novel and rigorous control-theoretic formalization of dialogue processes (DPs) in generative models, introducing definitions of reachability and controllability under a probabilistic and coarse-grained framework. The Monte Carlo–based estimation with PAC-style bounds is theoretically sound and well-motivated.
 However, several conceptual and practical issues remain: the definition of controllability in discrete, high-dimensional generative systems is somewhat ambiguous; key assumptions (boundedness, independence) are idealized . Overall, it is a theoretically interesting and well-written paper, but its practical implications and empirical validation need strengthening.

**Strengths:**

This paper proposes a control-theoretic framework to analyze and estimate the controllability and reachability of generative models (LLMs, T2IMs) viewed as black-box dynamical systems. The method provides PAC-style probabilistic guarantees for estimating controllable sets using Monte Carlo sampling, under minimal assumptions.
Strengths：
1.Theoretical originality – Introduces a formal and model-agnostic definition of controllability for generative systems, offering a new lens to study model behavior.
2.Sound probabilistic formulation – Provides PAC-based guarantees that make the estimation process interpretable and principled.
3.Clear empirical illustration – Demonstrates differences in controllability across models and tasks, supported by open-source implementation.

**Weaknesses:**

Conceptual Level: Ambiguity of Controllability Definition
The core concept of the paper, controllability, lacks a clear physical and semantic interpretation in the context of generative models. In classical control theory, controllability implies the existence of a set of inputs capable of steering the system from any initial state to any desired final state. However, for LLMs or T2I models: The boundaries of the state space X and input space U are semantically unclear;  Prompts or noise vectors are not strict control signals—they are discrete, symbolic inputs mapped nonlinearly to a latent space。Suggestion: Include an analysis (e.g., in the appendix) of the mapping between control inputs and semantic behaviors, clarifying the real-world meaning of the chosen control variables.

Experimental Depth and Analysis of Controllability Failures：
While the empirical examples, such as formality control and object counting, are illustrative, their interpretation remains relatively shallow. The study does not compare against simpler or alternative baselines, such as linear regression, RLHF-based control, or direct output editing. Moreover, the analysis of controllability failures—like overshoot phenomena—is limited to surface-level descriptions, without a deeper causal diagnosis.
Suggestion: Include qualitative case studies or detailed error analyses to clarify why certain models exhibit limited controllability and to provide more actionable insights.

**Questions:**

see Weaknesses

---

> ### Author Response · Authors · 2025-11-21
> **Response**
>
> Thanks for your constructive response! Please let us know if anything remains unclear.
>
> > Conceptual Level: Ambiguity of Controllability Definition The core concept of the paper, controllability, lacks a clear physical and semantic interpretation in the context of generative models.
>
> Section 3.1 is dedicated to setting up human-model dialogue as a control process, where we explicitly map all the components in the human-model dialogue to the components of a control system. In Section 3.2, we use the exact notion of controllability you mention (see Sontag, 1998) for the output attributes of generative models (see also https://en.wikipedia.org/wiki/Controllability#Output_controllability). We are happy to clarify any specific definition you find unclear.
>
> > Include an analysis (e.g., in the appendix) of the mapping between control inputs and semantic behaviors
>
> Note that we provide these experiments. For example, see Fig 2 (bottom) and Fig 3 for the exact mapping between control inputs and semantic behavior of the model. See Appendix A for these mappings for all other tasks (sentence length, even/odd number, object position, image saturation). Indeed, using the input-output mappings we are also able to report calibration metrics as described in Section 6.
>
> > Experimental Depth and Analysis of Controllability Failures： While the empirical examples, such as formality control and object counting, are illustrative, their interpretation remains relatively shallow. The study does not compare against simpler or alternative baselines, such as linear regression, RLHF-based control, or direct output editing. Moreover, the analysis of controllability failures—like overshoot phenomena—is limited to surface-level descriptions, without a deeper causal diagnosis. Suggestion: Include qualitative case studies or detailed error analyses to clarify why certain models exhibit limited controllability and to provide more actionable insights.
>
> Thanks for your suggestion. We added a qualitative case study on formality to Appendix A.4, finding that Qwen (the best performing model) adopts a strategy of mimicking the closest example, then adjusting the text incrementally in the direction of higher or lower formality according to feedback.
>
> Our work provides a framework to measure controllability with statistical guarantees, which has been an overlooked property of generative models. In this work we do not intend to design new controllers or compare to existing ones, but rather provide the right tools to practitioners so that they can diagnose their models with guarantees. Our experiments demonstrate that, even on simple tasks, controllability is fragile and should be analyzed. Practically, our framework also provides the one-to-one input-output mappings, as well as the individual trajectories in dialogue processes with $T>1$, which can be of great utility to practitioners to deepen the analysis and find root causes of controllability failures.
>
> However, we believe a thorough causal diagnosis of these failures is highly dependent on the controllability task and would require looking at model internals using tools in mechanistic interpretability. While important goals in themselves, controllability failure analysis and controller design are not our aim (see Section 3.2). Rather, our aim is to provide a rigorous tool for controllability failure *detection*.

---

> ### Author Response · Authors · 2025-11-28
> **Rebuttal Follow-Up**
>
> Dear Reviewer A4fa,
>
> Thank you for your thorough and constructive review. We have carefully considered your feedback and provided detailed responses in our rebuttal.
>
> To briefly summarize, we:
>
> * Clarified the conceptual definition of controllability by explicitly mapping human-model dialogue components to control systems and referencing established control theory definitions, while also highlighting existing figures that illustrate input-output mappings.
> * Addressed concerns about experimental depth and analysis of controllability failures by adding a qualitative case study (Appendix A.4). We also reiterated that our work provides a framework for measuring controllability with guarantees, offering tools for practitioners to diagnose failures, rather than designing or comparing control mechanisms.
>
> We hope these additions and clarifications have addressed your points. We would be grateful for any updated thoughts on our submission, particularly considering the extensive revisions and new evidence provided.
>
> Thank you for your time and expertise.
>
> Sincerely,
>
> Authors

---

### Author Response · Authors · 2025-11-21

Thanks all for your thoughtful and constructive reviews. We've incorporated your suggestions into the new pdf version (changes in blue). Please let us know if anything remains unclear!

---

### Meta-Review · Area_Chair_SF6W · 2026-01-06

**Summary:**

This paper proposes modeling user–model interaction as a control process and provides formal guarantees for estimating the controllable sets of generative models. Reviewers agree that the paper makes a meaningful and original contribution to the theoretical understanding and formal evaluation of controllability in generative systems. While reviewers note limitations in scalability to high-dimensional control problems and that the experimental evaluation is currently limited to prompt-based inputs, these concerns do not outweigh the paper’s strong theoretical contributions.

**Reviewer Concerns:**

### Addressed Concerns

- **Model size and controllability (Reviewer vGbb):**
  Reviewer vGbb noted that controllability fragility may primarily affect smaller models. In response, the authors added new experiments on the Qwen model series, demonstrating that controllability improves consistently as model size increases.

- **Theoretical clarity:**
  The authors have improved the clarity of the theoretical framework and strengthened its connection to classical control theory, addressing prior concerns about conceptual grounding.

### Outstanding Concerns

- **Scalability with large \(N\) (Reviewers ANgd, N39G):**
  As noted by Reviewers ANgd and N39G, the framework does not scale well when the number of discretized bins \(N\) becomes large, limiting applicability to high-dimensional control settings.

- **Restriction to prompt-based control (Reviewer ANgd):**
  Reviewer ANgd also highlights that the control signal is limited to text prompts, which can be sensitive and unreliable compared to alternative control mechanisms.

**Reviewer Scores:**

All reviewers provided positive scores for this paper, and the authors responded with detailed clarifications and additional discussions.

Reviewer **A4fa** requested greater clarity and a more in-depth analysis of model behavior. The authors addressed this by providing further explanation and adding new analyses on the Qwen models. As a result, this reviewer is likely to remain positive or potentially increase their score.

Reviewer **ANgd** raised concerns that control is restricted to text-based inputs and that the framework may not scale when \(N\) becomes large. The authors responded with clarifications and argued that \(N = 10\) already represents a practically large setting. This response is likely sufficient for the reviewer to maintain their current score.

Reviewer **N38G** questioned the behavior of the framework as \(N \to \infty\) and its generalization properties. The authors provided explanations addressing these concerns, and the reviewer is expected to maintain their score.

Reviewer **vGbb** requested experiments on larger models. The authors conducted additional experiments on the Qwen model series, addressing this concern directly. This reviewer is likely to remain positive or potentially raise their score.

---

### Decision · Program_Chairs · 2026-01-26

Accept (Poster)